# Gaussian Process Probes (GPP) for Uncertainty-Aware Probing

**Zi Wang**[*]
Google DeepMind

**Alexander Ku**[*]
Google DeepMind

**Jason Baldridge**
Google DeepMind

**Thomas L. Griffiths**
Princeton University

**Been Kim**
Google DeepMind

## Abstract

Understanding which concepts models can and cannot represent has been fundamental to many tasks: from effective and responsible use of models to detecting out of distribution data. We introduce Gaussian process probes (GPP), a unified and simple framework for probing and measuring uncertainty about concepts represented by models. As a Bayesian extension of linear probing methods, GPP asks what kind of distribution over classifiers (of concepts) is induced by the model. This distribution can be used to measure both what the model represents and how confident the probe is about what the model represents. GPP can be applied to any pre-trained model with vector representations of inputs (e.g., activations). It does not require access to training data, gradients, or the architecture. We validate GPP on datasets containing both synthetic and real images. Our experiments show it can (1) probe a model's representations of concepts even with a very small number of examples, (2) accurately measure both epistemic uncertainty (how confident the probe is) and aleatory uncertainty (how fuzzy the concepts are to the model), and (3) detect out of distribution data using those uncertainty measures as well as classic methods do. By using Gaussian processes to expand what probing can offer, GPP provides a data-efficient, versatile and uncertainty-aware tool for understanding and evaluating the capabilities of machine learning models.

## 1 Introduction

Deep learning models have become pervasive in a wide range of fields, yet they remain largely uninterpretable. As a result, gaining insight into which concepts deep learning models do or do not represent is of growing importance. This has led to the development of a family of methods known as "probes" [Alain and Bengio, 2016], which aim to understand the representations learned by these models. One common approach to probe the representations of a model is to train independent classifiers for each concept and measure their accuracy [Alain and Bengio, 2016, Kim et al., 2018, Belinkov, 2022]. For example, if the classifier returns $0.5$ probability (for the binary case) across a range of examples, we would reasonably conclude that the model does not represent the concept.

However, simply examining the output of a classifier may not adequately reflect the intricate nature of representing a concept. For example, we can think of two different scenarios that may result in the same classifier output. If you ask a person if an olive is a fruit, only about 60% will say yes (likewise 72% for avocado, 53% for pumpkin, and 40% for acorn) [McCloskey and Glucksberg, 1978]. This does not indicate that people have a limited representation of the *fruit* concept, but that

---

[*]Equal contribution. {wangzi, alexku, jasonbaldridge, beenkim}@google.com, tomg@princeton.edu. Our code can be found at https://github.com/google-research/gpax.

there is intrinsic fuzziness in the label. By contrast, if you show somebody an olive in a martini glass and tell them it is a new concept called "blicket," they may be 60% sure in judging whether an olive on a tree is a blicket (as opposed to "blicket" having a specialized meaning that applies to cocktails). In this second case, the uncertainty is because they do not have enough observations to be certain about what the new concept is – a common problem in word learning [Xu and Tenenbaum, 2007].

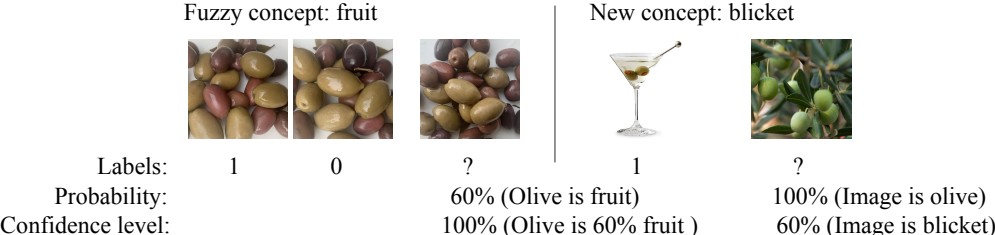

| | Fuzzy concept: fruit | | | New concept: blicket | |
|---|---|---|---|---|---|
| Labels: | 1 | 0 | ? | 1 | ? |
| Probability: | | | 60% (Olive is fruit) | | 100% (Image is olive) |
| Confidence level: | | | 100% (Olive is 60% fruit ) | | 60% (Image is blicket) |

We introduce a probabilistic approach based on Gaussian processes (GPs) [Rasmussen and Williams, 2006] that makes it possible to delineate different kinds of uncertainty in a model's concepts. Specifically, our goal is to probe a pre-trained model by constructing a probability distribution over possible binary classifiers of concepts induced by the model. Extracting this distribution allows the probe to produce two kinds of uncertainty measures – aleatory and epistemic [Fox and Ülkümen, 2011]. This allows us to distinguish cases where 1) there is fundamental fuzziness of the label (aleatory uncertainty) or 2) it does not have enough observations (epistemic uncertainty), whereas a binary classifier would predict equal probability for both and provide no further explanatory power.

Our framework also provides a natural building block for answering many downstream questions, such as detecting out-of-distribution (OOD) examples. Using the resulting uncertainty measures, we show that our method can correctly probe what a model represents, and that it achieves competitive performance in OOD detection. We validate this approach using both synthetic and real images.

## 2   Related work

Distinguishing aleatory and epistemic uncertainty has a long history in fields including cognitive science [Howell and Burnett, 1978, Fox and Ülkümen, 2011], probability theory [Jaynes, 2003], philosophy [Hacking, 2006] and more. Our work builds upon Fox and Ülkümen [2011], which introduced a framework for distinguishing the two dimensions of uncertainty from a human perspective.

In machine learning, Gaussian processes (GPs) and Bayesian neural nets (BNN) [] have enabled principled ways of predicting epistemic uncertainty, which led to better information-based decision making strategies [Houlsby et al., 2011, Hennig and Schuler, 2012, Wang and Jegelka, 2017, Gal et al., 2017, Wang et al., 2023]. Motivated by regression problems in reinforcement learning, Depeweg et al. [2018] decomposed uncertainty in BNN to a conditional entropy term as aleatory uncertainty and the mutual information between model weights and predictions as epistemic uncertainty. Our decomposition simplifies these quantities[2] and specializes them for classification. To our knowledge, aleatory uncertainty (fuzziness of concepts) has not been studied in the GP classification literature, and no previous work in the GP literature has been done to establish the correspondence between human perceptions of uncertainty and GP classification predictions.

Probing has been a popular approach to understand which concepts a model represents. There are mixed views of what probing should mean; either it is about estimating mutual information between representations and labels [Pimentel et al., 2020] or using minimum description length (MDL) [Voita and Titov, 2020] or representing the distance as squared Euclidean distance under a linear transformation [Hewitt and Manning, 2019], or using analysis with or without supervision [Saphra and Lopez, 2019, Wu et al., 2020]. Others considered different types of probing: conditional probing that compares two concepts [Hewitt et al., 2021], probing that asks where the task-relevant information "emerges" [Kunz and Kuhlmann, 2022] or which concepts are "used" [Lasri et al., 2022], and finally, probing/localizing that enables editing [Meng et al., 2022]. To the best of our knowledge, none of these probing methods offer measures of uncertainty.

---

[2]We define episteme as negative entropy, which has the same form as the mutual information $I(y; g) = H[y] - H[y \mid g]$ used by Depeweg et al. [2018], since equivalently, $I(y; g) = H[g] - H[g \mid y]$.

# 3 GPP: Gaussian process probes

We introduce Gaussian process probes (GPP), a simple probabilistic method for probing that naturally distinguishes aleatory and epistemic uncertainty. We formally define the problem of probabilistic probing in §3.1, followed by a review of the Beta Gaussian processes (GPs) that GPP builds upon in §3.2. In §3.3 and §3.4, we detail our method and how it measures uncertainty.

## 3.1 Problem formulation

Our goal is to probe a pre-trained model in a probabilistic manner by constructing a probability distribution over binary classifiers based on the model's vector representation of inputs (e.g., activations). Here, binary classifier refers to a function mapping a stimulus to the probability of a positive label.

**Assumption 1.** *The model, $\mathcal{M}$, contains basis functions that map from stimuli (e.g., images, texts) to a vector representation: $\phi : \mathcal{X} \mapsto \mathcal{A} \subset \mathbb{R}^d$, where $\mathcal{X}$ is the space of stimuli and $\mathcal{A}$ the space of $d$-dimensional real-valued vector representations.*

For example, basis functions $\phi$ can map from images to last-layer activations of a convolutional neural network (CNN). The probabilistic probe treats basis functions $\phi$ as a feed-forward black box, and does not require accessing the training data, gradients, or architecture used during model pre-training.

In order to define a stochastic process to describe a distribution over classifiers, the data generating process is assumed as follows.

**Assumption 2.** *There exists a stochastic process $\mathcal{G}(\theta)$ parameterized by $\theta$, and a classifier $h : \mathcal{A} \mapsto [0, 1]$ distributed according to $\mathcal{G}(\theta)$, such that the label $y \in \{0, 1\}$ for any vector representation $a \in \mathcal{A}$ is independently distributed according to a Bernoulli distribution with parameter $h(a)$; i.e., $p(y \mid a, h) = h(a)^y (1 - h(a))^{1-y}$.*

Now the posterior inference of the stochastic process (in our case, Beta GP) requires conditioning on observations, and the stochastic process measures uncertainty for a set of queries. We denote a set of observations as $D$, which consists of representation-label pairs, i.e., $D = \{(\phi(x_i), y_i)\}_{i=1}^{|D|}$; here, label $y_i$ is either 0 or 1. The queries, $\boldsymbol{a}_q = [\phi(x_j)]_{j=1}^{|\boldsymbol{a}_q|} \in \mathbb{R}^{d \times |\boldsymbol{a}_q|}$, are vector representations of a set of stimuli whose labels are to be predicted together with uncertainty measures.

Note that there are no additional assumptions about this probing dataset, e.g., the distributions of stimuli in observations or queries. We use GPP for OOD detection, as shown later. Unlike existing probes (e.g., [Hewitt et al., 2021, Xu et al., 2020]), we do not consider a stimulus and its vector representation as random variables with some underlying distributions. Instead, viewing them as deterministic allows us to probe with any stimuli (with potentially any underlying distributions).

## 3.2 Background: Beta Gaussian processes

GPP uses a Beta GP to define a prior distribution over classifiers. The Beta GP is a special case of Dirichlet-based GPs [Milios et al., 2018], which have been shown to either outperform or achieve similar performance than classic GP classification approximations [Rasmussen and Williams, 2006]. For each representation $a \in \mathcal{A}$, Beta GPs assume the binary label $y$ is generated by sampling $y \sim \text{Bernoulli}(g(a))$ where the Bernoulli parameter $g(a) \sim \text{Beta}(\alpha(a), \beta(a))$. The Beta variable $g(a)$ can be written as two independent Gamma variables:

$$g(a) = \frac{t_\alpha(a)}{t_\alpha(a) + t_\beta(a)}, \ t_\alpha(a) \sim \text{Gamma}(\alpha(a), 1) \ \text{and} \ t_\beta(a) \sim \text{Gamma}(\beta(a), 1),$$

where $t_\alpha(a)$ and $t_\beta(a)$ are independent. The Gamma distributions are then approximated with Log-normal distributions via moment matching. That means,

$$f_\alpha(a) = \log(t_\alpha(a)) \sim \mathcal{N}(\log(\alpha(a)) - \frac{v_\alpha}{2}, v_\alpha), \ \text{where} \ v_\alpha = \log(\frac{1}{\alpha(a)} + 1), \tag{1}$$

and the same for $f_\beta$. Beta GPs use two GPs to model the two latent functions $f_\alpha$ and $f_\beta$,

$$g = \frac{1}{1 + e^{-f}}, \ \text{where} \ f = f_\alpha - f_\beta, f_\alpha \sim \mathcal{GP}(\mu, k), f_\beta \sim \mathcal{GP}(\mu, k), f_\alpha \perp\!\!\!\perp f_\beta. \tag{2}$$

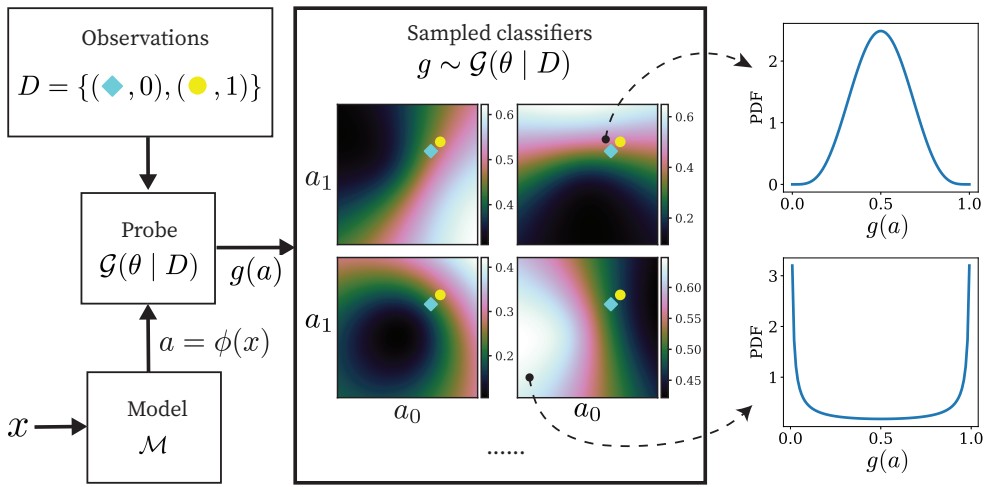

Figure 1: Illustrative example of probabilistic probing: The model of interest $\mathcal{M}$ transforms a stimulus $x$ to its vector representation $a = \phi(x) \in \mathbb{R}^2$. The probe, $\mathcal{G}(\theta \mid D)$, a distribution parameterized by $\theta$ over possible classifiers (mapping from representations to predicted probabilities) conditional on the observation set $D$. Suppose $D$ contains two datapoints. Sampled classifiers $g$ from $\mathcal{G}(\theta \mid D)$ are depicted in the middle box. The aggregation of predictions from the sampled classifiers produces the distribution for $g(a)$; $g(a)$ is the predicted probability of any representation $a$. Notice that the two data points in $D$ are located close to each other, yet has opposite labels, indicating fuzziness in labels around this region. Imagine a new query data $a$. In the top right-most, we see that $a$ close to $D$ produces a distribution that is centered around $g(a) = 0.5$. If the representation $a$ is far away from $D$ (bottom right-most), the probe can have higher epistemic uncertainty due to lack of information, and the predicted probability follows the prior distribution (in this case, the prior has two modes, one at $g(a) = 0$ and one at $g(a) = 1$). Note that both distributions have an expectation at $g(a) = 0.5$, for drastically different reasons: fuzzy label v.s. lack of knowledge.

Note that the latent function $f$ in Eq. 2 is still a GP given that it is the sum of two independent GPs. We denote the Beta GP as $\mathcal{G}(\theta)$, where $\theta = (\mu, k)$. While we focus on probing as binary classification in this work, it is straightforward to extend GPP to multi-class classification by switching the Beta prior to a Dirichlet prior [Milios et al., 2018] (i.e., using more latent functions than just $f_\alpha$ and $f_\beta$).

### 3.3 Adapting Beta GPs for GPP

There are two important details in Beta GPs that require special attention: how to set the prior and how to approximate the posterior of classifier $g$. This requires setting mean and kernel functions $\mu, k$ in the prior $\mathcal{GP}(\mu, k)$ of the latent functions $f_\alpha$ and $f_\beta$, and computing their posterior $f_\alpha, f_\beta \mid D$.

We adapt these aspects of Beta GPs for probing settings, that also enables users to incorporate the characteristics of their end-tasks by setting intuitive hyperparameters. See more explanations and illustrations in the Appendix.

#### 3.3.1 Setting the prior

First, we need to make sure the prior distribution approximated by GPs in our GPP matches with the Beta prior. Without loss of generality, we assume that for any $a \in \mathcal{A}$, the prior distribution for $g(a)$ is Beta$(\epsilon, \epsilon)$ for some $\epsilon > 0$. This hyperparameter $\epsilon$ reflects the characteristics of the task at hand; the probability of a positive label might be bi-modal with density centered at both 0 and 1 (e.g., label is not fuzzy) ($\epsilon < 1$), or uniformly distribution in $[0, 1]$ ($\epsilon = 1$), or centered at 0.5 ($\epsilon > 1$).

To match the Beta prior, the prior for both $f_\alpha(a)$ and $f_\beta(a)$ has to be $\mathcal{N}(\log(\epsilon) - \frac{v}{2}, v)$, where $v = \log(\frac{1}{\epsilon} + 1)$. To match this normal distribution, we set the mean function to a constant $\mu(a) = \log(\epsilon) - \frac{v}{2}$, and constrain the kernel function to satisfy $k(a, a) = v$.

### 3.3.2 Posterior inference

This formulation offers a closed-form solution for posterior inference of the GP. Namely, any $y_i = 1$ in $D = \{(a_i, y_i)\}_{i=1}^{|D|}$ results in updating Beta parameters of $g(a_i)$ to be $\text{Beta}(\epsilon + s, \epsilon)$ and any $y_i = 0$ results in $\text{Beta}(\epsilon, \epsilon + s)$, for some $s \geq 1$. This hyperparameter $s$ describes how much "strength" (i.e., weight of the observation) we add to the Beta posterior. To match the GP posterior to the Beta posterior, we synthetically add observations with heteroscedastic noise to functions $f_\alpha$ and $f_\beta$. If $y_i = 1$, the observation for $f_\alpha$ is $(a_i, \log(\epsilon + s) - \frac{v'}{2})$ with noise $\mathcal{N}(0, v')$ where $v' = \log(\frac{1}{\epsilon + s} + 1)$, and the observation for $f_\beta$ is $(a_i, \log(\epsilon) - \frac{v''}{2})$ with noise $\mathcal{N}(0, v'')$ where $v'' = \log(\frac{1}{\epsilon} + 1)$. And vice versa for $y_i = 0$. The posterior of a GP remains a GP with closed-form updates to its mean and kernel function. Once we have the posterior of latent functions $f_\alpha \mid D$ and $f_\beta \mid D$, obtaining the posterior of classifier $g$ is trivial by updating the GP priors in Eq. 2 to the GP posteriors.

### 3.3.3 The cosine kernel

The choice of kernel function is important for GPP as it directly determines the characteristics of the distribution of latent functions, which translates to the distribution of classifiers that GPP uses. We define a cosine kernel as:

$$k(a, a') = v \frac{a^\top a' + 1}{(\|a\| + 1)^{\frac{1}{2}}(\|a'\| + 1)^{\frac{1}{2}}}, \text{ where } v = \log(\frac{1}{\epsilon} + 1). \tag{3}$$

Note that multiplying the constant $v$ is the constraint we have from setting the prior for the Beta GP. Using the cosine kernel in the GP is equivalent to defining a distribution over linear latent functions in an augmented space of the original representation space $\mathcal{A}$, i.e.,

$$f_\alpha(a) = W_\alpha^\top \psi(a) + \mu(a), \text{ where } \psi(a) = \frac{\sqrt{v}}{\|a\| + 1} \begin{bmatrix} a \\ 1 \end{bmatrix} \in \mathbb{R}^{d+1} \text{ and } W_\alpha \sim \mathcal{N}(0, I). \tag{4}$$

And similarly for $f_\beta$ with weights $W_\beta$. Here $\mu$ is the mean function of the Beta GP. Note that we have the closed-form posterior distribution for the weights, which remains a multivariate Gaussian distribution. See Appendix for more details.

Why is the kernel defined in this way? During the pre-training stage of deep learning models, a weighted sum of the activations and a bias term are used to construct neurons of the next layer. Following Wang et al. [2023], we can view each next layer neuron as independent samples of classifiers from the same distribution, described by a Beta GP in this work. This means classifiers from the Beta GP should all consider a bias term. Hence we augment the activations by an additional dimension with value fixed at 1 (Eq. 4). Additionally, we normalize the augmented activations, $\psi(a)$, such that $k(a, a) = v$ (the constraint from setting the prior in §3.2) holds for all $a \in \mathcal{A}$ (Eq. 3).

We can interpret the cosine kernel as a scaled cosine of the angle between two vectors (standard cosine similarity) in an augmented representation space described by activations and biases.

## 3.4 Uncertainty measures

The probabilistic probing formulation makes it possible to compute a posterior distribution over labels for queries. Building on terms from Fox and Ülkümen [2011], we define uncertainty measures that identify the epistemic and aleatory uncertainty of the posterior prediction $g(a)$ where $g = \frac{1}{1 + e^{-f}} \sim \mathcal{G}(\theta \mid D), \forall a \in \mathcal{A}$. Let the posterior of the latent function be $f(a) \mid D \sim \mathcal{N}(\mu_D(a), k_D(a))$.

- *Episteme* $:= -\mathbb{H}[g(a)]$, the negative entropy of the distribution of $g(a)$, which describes the amount of knowledge the probe has about the label of stimulus $a$. In GPP, we have $\mathbb{H}[g(a)] = \mathbb{H}[f(a)] - \mu_D(a) - 2\mathbb{E}[\log(1 + e^{-f(a)})]$ and $\mathbb{H}[f(a)] = \frac{1}{2}\log(2\pi e k_D(a))$.
- *Alea* $:= \mathbb{H}[y \mid g(a)] = \mathbb{E}[-g(a)\log(g(a)) - (1 - g(a))\log(1 - g(a))]$, the expected entropy of the conditional distribution $p(y \mid g(a))$, where $y$ is the label of stimulus $a$. Higher alea corresponds to more fuzziness in the label of $a$.
- *Judged probability* $:= \mathbb{E}[g(a)] = \mathbb{E}[\frac{1}{1 - e^{-f(a)}}]$, which is the expected probability (judged by the probe) that the label of stimulus $a$ is positive.

Figure 2 shows the relationship of episteme, alea and judged probability for a rational agent (white region, based on [Fox and Ülkümen, 2011]).

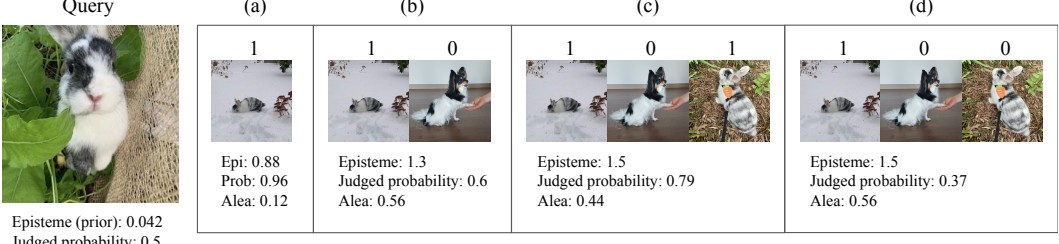

| Query | (a) | (b) | (c) | (d) |

Episteme (prior): 0.042
Judged probability: 0.5
Alea: 0.44

(a) Epi: 0.88 / Prob: 0.96 / Alea: 0.12

(b) Episteme: 1.3 / Judged probability: 0.6 / Alea: 0.56

(c) Episteme: 1.5 / Judged probability: 0.79 / Alea: 0.44

(d) Episteme: 1.5 / Judged probability: 0.37 / Alea: 0.56

Observations

Figure 3: How episteme, judged probability and alea of GPP changes as more observations are given to the CoCa model [Yu et al., 2022]. As observations are added, episteme increases while judged probability fluctuates. (a) With 1 positive observation, the judged probability for the query is 0.96, showing that the query and an the observation are very close in representation space. (b) With 1 positive (rabbit) and 1 negative (dog) example, the judged probability lowers to 0.6, showing that the CoCa model would also put rabbits and dogs close in representations, but the queried rabbit is still closer to the observed rabbit. (c) With 1 more positive example of a rabbit, judged probability increases together with episteme. (d) If we set the 3rd observation in (c) to be negative, GPP gets more evidence that the query is probably negative and the judged probability decreases to 0.37.

Low episteme means we are "not sure" and high episteme that we are "highly confident" about the underlying probability. Alea, on the other hand, increases with the intrinsic randomness of the true label. The judged probability that the stimulus's label is positive reflects both episteme and alea. With low episteme, the judged probability should be closer to 0.5, meaning the probe does not have enough information to draw a plausible conclusion. It would be irrational (grey areas in Figure 2) for a person to believe something will definitely happen even if there is no evidence. Analogously, it is irrational for a probe to produce probabilities close to 0 or 1 while having low episteme. By accumulating evidence (increasing episteme), the judged prob-

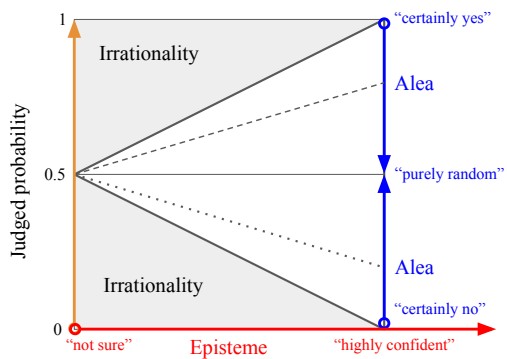

Figure 2: Interaction of alea and episteme (recreation of Figure 1 from Fox and Ülkümen [2011]).

ability can move closer and closer to the ground truth (moving to the right in the figure). An accurate probe should produce a judged probability that matches the ground truth fuzziness of a label only if episteme is high.

Figure 3 is a real-world example (using a CoCa model [Yu et al., 2022] and natural images, shown in the figure, as query and observations) of how our measures of uncertainty change as we increase a small number of positive and negative observations.

Table 1 presents the results of applying GPP to the concepts involving olives introduced in §1. The judged probabilities from GPP match human judgments [McCloskey and Glucksberg, 1978]: olives have a 57% chance of being fruit with a high level of confidence, and GPP is very uncertain about whether olives on a tree are blickets.

Table 1: GPP predictions for olives in §1.

| Concept | Judged prob. | Episteme | Alea |
|---------|--------------|----------|------|
| Fruit | 57% | 2.3 | 0.63 |
| Blicket | 91% | 0.49 | 0.22 |

## 4 Validating GPP

In addition to probing concepts, GPP predicts aleatory and epistemic uncertainty, which allows us to determine the fuzziness of a concept or detect out-of-domain (OOD) stimuli.

We use 3D Shapes [Burgess and Kim, 2018] and construct 3 datasets based on its concept ontology to validate these uses of uncertainty measures. In these 3 settings, we can control the ground truth level of fuzziness by artificially injecting noise to labels (§4.2, §4.3). In §4.5, we evaluate the OOD performance of GPP and competitive baselines on 3D Shapes as well as 10 datasets defined by coarse-grained labels of the ImageNet dataset [Russakovsky et al., 2015].

## 4.1 Experiment setups for probing

We evaluate GPP's ability to probe using the 3D Shapes dataset [Burgess and Kim, 2018], which contains images generated from 6 ground truth independent primitives: 10 floor colors, 10 wall colors, 10 object colors, 8 scales, 4 shapes and 15 orientations of the shapes, as shown below.

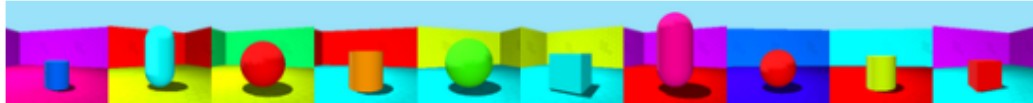

**Datasets and models.**  We construct a 2-level concept ontology (see Appendix for visualization) consisting of a disjunctive level where colors are binarized to warm/cool and scale to small/large, and a conjunctive level that combines shapes, binarized colors and binarized scales.

We train three CNN models on labels generated from this ontology: 1) $\mathcal{M}.1$ is trained on 64 labels (binarized scale, floor, wall object color x 4 shapes), 2) $\mathcal{M}.2$ is trained on 8 labels (4 shapes x binarized scale) 3) $\mathcal{M}.3$ is trained on 8 labels (binary floor, wall and object color).

We define two probing tasks to investigate each model: 1) task $\mathcal{P}.1$: binarized colors (floor, wall, object) and 2) task $\mathcal{P}.1$: binarized scales and 4 shapes.

**Baseline descriptions.**  The baselines include 1) LP: linear probes [Alain and Bengio, 2016]; 2) SVM: SVM probes (a baseline from Kim et al. [2018]); 3) LPE: linear probes ensembled via bootstrap [Kim et al., 2018]. For each linear probe in LPE, we train a logistic regression classifier on a dataset sampled from the original set of observations with replacement. The size of the dataset is the same as the number of observations. Each ensemble has 100 linear probes. LPE can also describe a distribution over classifiers and be evaluated as probabilistic probes, making this a strong baseline. For GPP, we use $\text{Beta}(0.1, 0.1)$ as the prior and set strength to be 5.

## 4.2 Validating GPP as a probe

To compare with baseline methods, we use AUROC to evaluate the deterministic classifier defined by the judged probability $\mathbb{E}[g(a)], \forall a \in \boldsymbol{a}_q$ (defined in §3.4). Queries are sampled disjointly from the training data and observations. Figure 4 shows that GPP performs competitively against LP and SVM. GPP and LP perform comparable with respect to AUROC and sample efficiency (needing as few as 10 observations for probing concepts in task $\mathcal{P}.1$).

One may first guess that $\mathcal{M}.3$ is the best model that represents color, since it was explicitly tasked to distinguish color. However, our probing results reveal what is obvious in hindsight: while $\mathcal{M}.2$ is only tasked to distinguish shapes and scales, separating where the shapes are requires detecting color (i.e., the only cue to separate shape from background is color). This is shown in $\mathcal{M}.2$'s high AUROC for $\mathcal{P}.1$ in Figure 4. $\mathcal{M}.3$ is only tasked to distinguish colors, so performance on $\mathcal{P}.2$ remains low.

## 4.3 Estimating concept fuzziness

What happens when the concepts we are trying to probe are inherently fuzzy? This is commonly the case in the real world, where situations and the language used to describe them contain extensive ambiguity. We again focus on $\mathcal{M}.1$ and concepts in task $\mathcal{P}.1$. We control the fuzziness of concepts in $\mathcal{P}.1$ by randomly flipping positive labels to negative (but not vice versa) in the observations. We test ground truth label probabilities of $0.25, 0.5, 0.75$ and $1$, where $1$ indicates no fuzziness. To measure GPP's alignment with ground truth label probabilities, we use the Pearson correlation coefficient between ground truth label probabilities and judged probabilities for all queries.

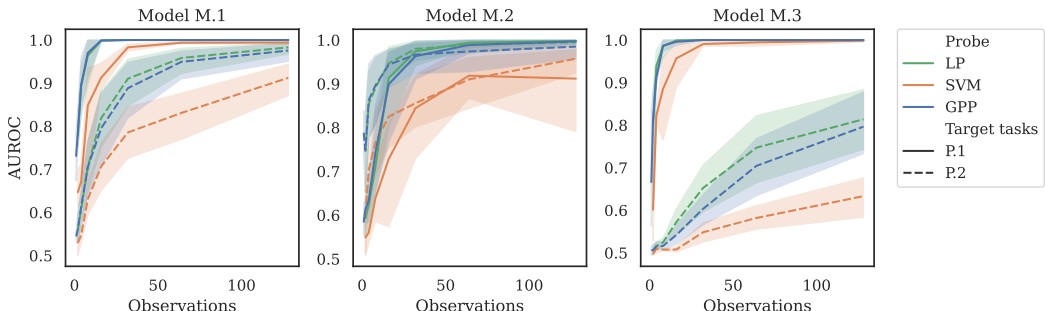

Figure 4: AUROC results for probing $\mathcal{M}.1$, $\mathcal{M}.2$ and $\mathcal{M}.3$ with $\mathcal{P}.1$ and $\mathcal{P}.2$ tasks. $\mathcal{M}.1$ and $\mathcal{M}.3$ are trained on color-related tasks and all probes obtain better performance on color concepts $\mathcal{P}.1$ than geometry concepts $\mathcal{P}.2$. The probing results reveal something obvious in hindsight; while $\mathcal{M}.2$ is trained on geometry-related tasks, it needed color to separate the object from the background color in order to detect geometry, resulting in significant representation of color concepts, $\mathcal{P}.1$.

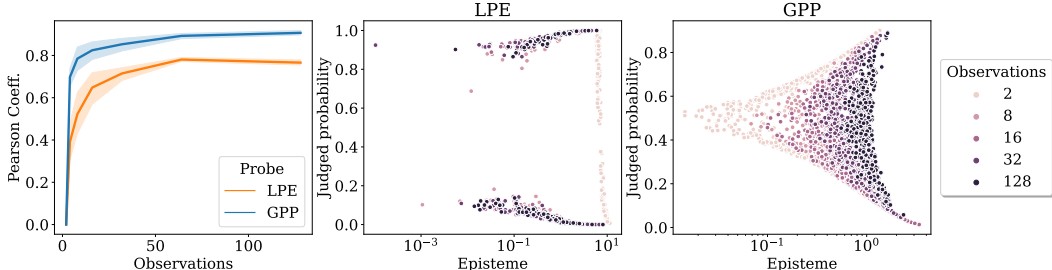

Figure 5: Left: Pearson correlation coefficient between ground truth label probabilities and judged probabilities. GPP obtains a much higher correlation than LPE, showing that GPP is better at detecting fuzzy concepts. Middle and Right: Relationship between judged probability and episteme of positive labels using LPE and GPP when ground truth label probability is 0.5. LPE fails to detect fuzziness and assigns extreme judged probabilities to queries for 8 or more observations. With 2 observations, LPE has higher episteme than with more observations. This behavior is irrational [Fox and Ülkümen, 2011]: fewer observations should indicate lower episteme, and with low episteme high judged probability indicates LPE is being confidently ignorant (no knowledge but believes something will definitely happen). On the contrary, GPP shows rational behavior: with high episteme, the mass of GPP's judged probability predictions centers at 0.5, which aligns with the ground truth.

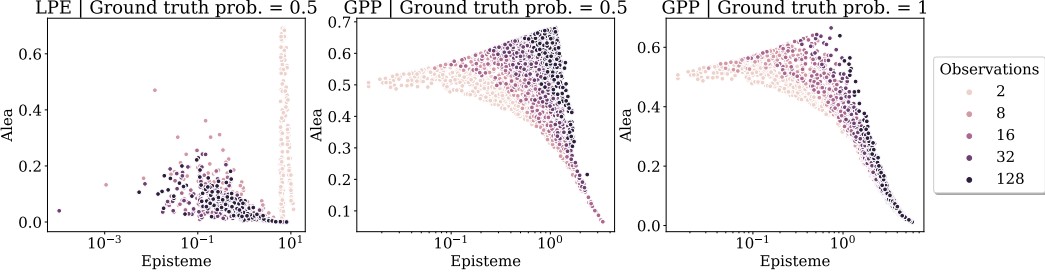

Figure 6: Alea predictions as a function of episteme for LPE and GPP. When ground truth label probability is 0.5 (Left and Middle), LPE predicts low alea for observations larger than 2, which does not agree with the ground truth alea (0.69). GPP shows much more rational predictions. Comparing Middle and Right, GPP predicts higher alea with more observations for ground truth label probability 0.5 (the corresponding alea is 0.69), and its alea predictions converge to 0 (ground truth alea is 0) for ground truth label probability 1.

Figure 5 shows that GPP can accurately probe fuzzy concepts by predicting both judged probability (expected level of fuzziness) and episteme (how sure it is about the judged probability). GPP achieves a higher Pearson correlation coefficient than LPE across all sizes of observations. In particular, the relationship between judged probabilities and episteme coincides with the two-dimensional framework for characterizing uncertainty from Fox and Ülkümen [2011] (see Fig. 2), showing the validity of GPP as a rational method to probe fuzziness.

Figure 5 (Right) confirms that the judged probability alone cannot be used to distinguish whether concept is fuzzy or the stimuli are far away: in both cases, judged probabilities can be $0.5$. Hence, it is important to remember that judged probability is only accurate with enough observations. We illustrate this point from another angle in Figure 6, which shows the relation between alea and episteme predicted by LPE, GPP with ground truth label probability $0.5$ or $1$. When ground truth label probability is $0.5$, GPP can have low alea under low episteme (same effect as Figure 1), but with more episteme, GPP puts more mass of predictions for high alea because the ground truth alea is high. LPE does not show this kind of rational behaviors. When ground truth label probability is $1$, more mass of alea predicted by GPP converges to $0$ as episteme increases, since the ground truth alea is $0$.

### 4.4 Experiment setups for OOD detection

When learning new concepts, it is natural for people to be able to say "I'm not sure since I haven't learned it yet". For probes, the corresponding ability is to perform out-of-distribution (OOD) detection, which identifies datapoints that are different from the observed training data. GPP naturally performs OOD detection using episteme predictions. With an uninformative prior for the labels,[3] high episteme indicates in-domain (ID), while low episteme indicates OOD.

**Method descriptions.** For GPP, we use the negative latent variance of posterior function $f$ (Eq. 2) as a proxy of episteme for computational efficiency. It is easy to show that the entropy of classifier $g$ is upper bounded by the entropy of latent function $f$, which is fully determined by its posterior variance (more details in Appendix). In practice, we found that using the negative latent variance as OOD detection scores for GPP is robust to the level of informativeness of the Beta prior, because the latent GPs are agnostic with respect to the logistic transformation in Eq. 2.

As for baselines, we compare against MSP (maximum predicted softmax probabilities using LP) [Hendrycks and Gimpel, 2016], Maha (negative Mahalanobis distance-based score) [Lee et al., 2018], NN (deep nearest neighbors) [Sun et al., 2022] and LPE. Recall that LPE uses bootstrapping to create an ensemble of linear probes. Each linear probe in LPE can be viewed as *i.i.d.* samples from an underlying distribution over classifiers (i.e., $g$ in Figure 1). Similar to approximating metrics with samples in §3.4, we can estimate the negative variance of predictions from the set of linear probes, and use it as the OOD detection score for LPE.

As shown in the extensive analyses of OOD detection methods and tasks in Appendix E of Tran et al. [2022], Maha and LPE (LPE is equivalent to their "Entropy" method for ensembles) achieved top performance, surpassing more recent proposals from Ren et al. [2021].

**Datasets and models.** We generate a set of queries from task $\mathcal{P}.1$ of 3D Shapes that include 1024 ID images and 1024 OOD images (uniform random noise), and see if OOD queries can be detected using the methods above based on representations from model $\mathcal{M}.1$.

We also evaluate the OOD detection performance of all methods on real world datasets and models. Queries and observations in the real world datasets are sampled disjointly from the validation split of the ImageNet dataset [Russakovsky et al., 2015]. We make 10 sets of $Ds$ using 10 binary classification tasks defined by supersets of ImageNet classes. Query points consist of 128 images from the classes the probe has observed and 128 OOD images (uniform random noise). The ID examples are images that come from the same distribution that the probe observes. For example, consider a probe trained to classify "all dogs" vs "all cats"; images of dogs and cats would be ID, whereas random noise would be OOD. The models we used are the pre-trained ResNet-50 [He et al., 2016] and CoCa-Base [Yu et al., 2022]. See more details in Appendix.

---

[3]For example, Beta$(0.1, 0.1)$ is a more informative prior than Beta$(1, 1)$, because the former means the classifier outputs either $0$ or $1$ with high probability, while the latter puts a uniform distribution over any real numbers in $[0, 1]$. See Figure 9 for visualization of the PDF of Beta distributions.

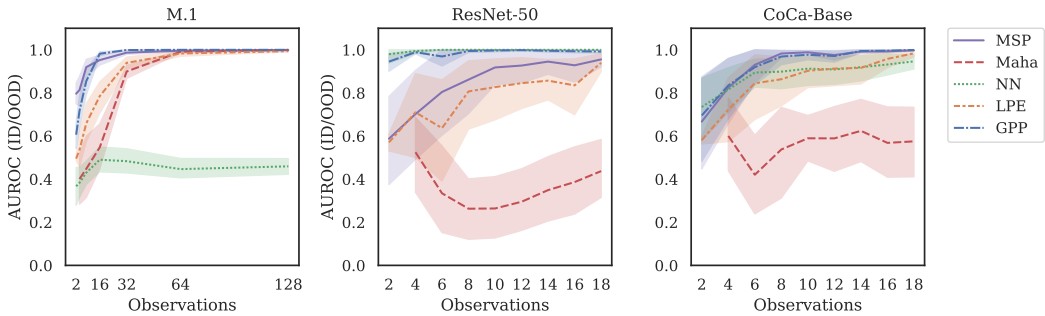

Figure 7: AUROC for ID/OOD detection with an increasing number of observations for task $\mathcal{P}.1$ of 3D Shapes (Left) and real-world datasets constructed from ImageNet (Middle and Right), showing GPP's comparable performance and data efficiency in OOD detection.

## 4.5 Results for OOD detection

Figure 7 shows the competitive performance of GPP as an OOD detector for all the OOD detection tasks we constructed. While GPP might not be the best OOD detection approach for each of the tasks, GPP performed consistently well and on par with the best benchmark method for each task. This means, while GPP is mainly a probing method, it is also competitive for OOD detection.

Note that the effective term in GPP's negative latent posterior variance, $k(a, \boldsymbol{a}_o)K^{-1}k(\boldsymbol{a}_o, a)$ (here $\boldsymbol{a}_o$ is the concatenated observed activations and $K = k(\boldsymbol{a}_o, \boldsymbol{a}_o) \in \mathbb{R}^{O \times O}$), looks very similar to Mahalanobis distance, $(a - \hat{u}_c)^\top \hat{\Sigma}^{-1}(a - \hat{u}_c)$ (Eq. 2 from Lee et al. [2018], where $\hat{\Sigma} \in \mathbb{R}^{d \times d}$; $d$ is the dimension of activations). However, GPP's variance measures the similarity between function values but Mahalanobis distance measures the similarity between activations. Also note that the Gaussian estimated for Mahalanobis distance is degenerate for $d > O$, and so the distance is still computed on a $O$-dimensional manifold; this is possibly why the Maha method shows inferior performance.

## 5 Conclusion

We introduced GPP, an uncertainty-aware probing framework that uses Gaussian processes to measure both the aleatory uncertainty intrinsic to the concept represented by a model and the epistemic uncertainty reflecting the probe's confidence about its judgement. These uncertainty measurements also equip GPP with the ability to detect OOD stimuli. Empirically, we verified the strong performance of GPP for data-efficient probing that can correctly surface fuzziness in concepts and OOD detection which shows GPP can also surface epistemic uncertainty (i.e., knows that it does not know). These results illustrate how distinguishing between different forms of uncertainty can be beneficial both for deepening our theoretical understanding of models and for practical applications.

**Limitations.** We have validated our approach on both synthetic and real-world datasets, and demonstrated its use for OOD detection, but there are a much wider range of models, datasets, and applications that can be explored. Fully evaluating the potential of this approach and achieving the greatest impact will require pursuing this investigation, which we hope to do in future work. In particular, we plan to explore active few-shot learning and distributionally robust probing. Finally, GPP can only be used with models for which activations are made available, which may present an obstacle to use with certain commercial machine learning systems.

**Broader impact.** Being able to understand what models can and cannot represent has huge implications. Not only does it impact important downstream tasks, such as OOD detection, but it ultimately aids human-machine communication and collaboration. While a (potentially) superficial interaction (e.g., with a chatbot) is already possible, we also know models hallucinate [Ji et al., 2023], leaving people unsure about the validity/faithfulness of this interaction. Work such as ours ultimately aims to help humans to better use and develop models that aligns well with our concepts, as well as potentially learning new concepts from machines.

## Acknowledgments and Disclosure of Funding

We thank Fei Sha for constructive discussions on aleatory uncertainty in classification. We also thank Geoff Pleiss, Alexander Terenin, Ben Adlam, Jasper Snoek and Zackary Nado for helpful conversations on Gaussian process classification. Images of "blicket" from https://www.iwawine.com/Images/riedel-vinum-martini-glasses_10.jpg and https://libreshot.com/wp-content/uploads/2017/12/green-olives-on-the-tree.jpg (by Martin Vorel).

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

Table 2: Notations for GPP.

| Terminology | Symbol | Meaning |
|---|---|---|
| Model | $\mathcal{M}$ | The model to probe. |
| Stimuli space | $\mathcal{X}$ | The space of stimuli. |
| Representation space | $\mathcal{A} \subseteq \mathbb{R}^d$ | The space of vector representations of $\mathcal{M}$. |
| Basis functions | $\phi$ | The function (contained in $\mathcal{M}$) mapping from $\mathcal{X}$ to $\mathcal{A}$. |
| Vector representation | $\phi(x)$ | The vector representation of a stimulus $x \in \mathcal{X}$. |
| | $\alpha$ | $\alpha : \mathcal{A} \mapsto \mathbb{R}^+$, mapping to the 1st parameter of a Beta distribution. |
| | $\beta$ | $\beta : \mathcal{A} \mapsto \mathbb{R}^+$, mapping to the 2nd parameter of a Beta distribution. |
| | $t_\alpha$ | A random function such that $t_\alpha(a) \sim \mathrm{Gamma}(\alpha(a), 1)$. |
| | $t_\beta$ | A random function such that $t_\beta(a) \sim \mathrm{Gamma}(\beta(a), 1)$. |
| Beta distribution | $\mathrm{Beta}(\alpha(a), \beta(a))$ | The Beta distribution for $g(a)$, $\forall a \in \mathcal{A}$. |
| Mean function | $\mu$ | $\mu : \mathcal{A} \mapsto \mathbb{R}$. |
| Kernel function | $k$ | $k : \mathcal{A} \times \mathcal{A} \mapsto \mathbb{R}$. |
| GP | $\mathcal{GP}(\mu, k)$ | A GP with mean function $\mu$ and kernel function $k$. |
| | $f_\alpha \sim \mathcal{GP}(\mu, k)$ | $f_\alpha : \mathcal{A} \mapsto \mathbb{R}$, the (latent) function for approximating $t_\alpha$. |
| | $f_\beta \sim \mathcal{GP}(\mu, k)$ | $f_\beta : \mathcal{A} \mapsto \mathbb{R}$, the (latent) function for approximating $t_\beta$. |
| | $\theta$ | Parameter for the Beta GP in GPP, $\theta = (\mu, k)$. |
| Beta GP | $\mathcal{G}(\theta)$ | A distribution over functions mapping from $\mathcal{A}$ to $[0, 1]$. |
| Classifier | $g \sim \mathcal{G}(\theta)$ | $g = \frac{1}{1+e^{-f}} : \mathcal{A} \mapsto [0, 1]$, a random function. |
| Observations | $D$ | $D = \{(\phi(x_i), y_i)\}_{i=1}^{|D|}, x_i \in \mathcal{X}, y_i \in \{0, 1\}$. |
| Beta GP posterior | $\mathcal{G}(\theta \mid D)$ | The Beta GP conditional on observations $D$. |
| Queries | $\boldsymbol{a}_q$ | $\boldsymbol{a}_q = [\phi(x_j)]_{j=1}^{|\boldsymbol{a}_q|} \in \mathbb{R}^{d \times |\boldsymbol{a}_q|}$. |
| Predicted probabilities | $g(\boldsymbol{a}_q)$ | $g(\boldsymbol{a}_q) = [g(a)]_{a \in \boldsymbol{a}_q}$. |

# A   Notation

In Table 2, we include the main notation used in this paper.

# B   Details on the Beta GP in GPP

As shown in § 3.2 and §3.3, with observations $D = \{(a_i, y_i)\}_{i=1}^{|D|}$, the posterior of a Beta GP can be written as the transformed version of a GP, i.e.,

$$g = \frac{1}{1 + e^{-f}} \sim \mathcal{G}(\theta \mid D), \text{ where } f \sim \mathcal{GP}(\mu_D, k_D).$$

We show how to obtain the mean and kernel functions $\mu_D$, $k_D$ in §B.1, as well as the posterior of weights in §B.2 for the cosine kernel. The behavior of GPP relies on two hyperparameters, $\epsilon$ and $s$, and we explain what they are and how to set them in §B.3. In §B.4, we analyze and bound the episteme of GPP.

As a reference, Figure 8 shows the graphical model of GPP.

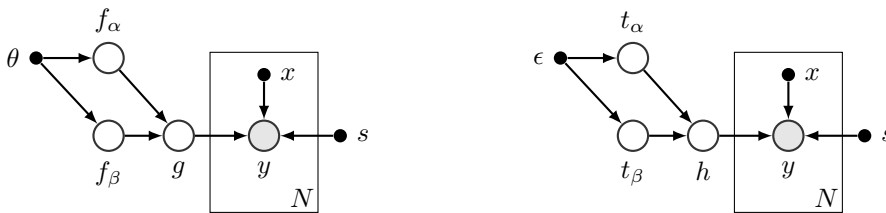

Figure 8: Left: Graphical model of the Beta GP in GPP with $N$ observations. Right: The stochastic process that a Beta GP approximates, where $h = \frac{t_\alpha}{t_\alpha + t_\beta} \sim \text{Beta}(\epsilon, \epsilon), t_\alpha \sim \text{Gamma}(\epsilon, 1), t_\beta \sim \text{Gamma}(\epsilon, 1)$, and $t_\alpha \perp\!\!\!\perp t_\beta$. In Beta GP, $g = \frac{e^{f_\alpha}}{e^{f_\alpha} + e^{f_\beta}}$, where $f_\alpha$ approximates $\log t_\alpha$, and $f_\beta$ approximates $\log t_\beta$. Hyperparameter $s$ is the weight for each observation.

### B.1 Posterior inference (extension of §3.3.2)

Without loss of generality, we write observations as a union of a dataset (of size $n$) with the positive labels only and a dataset (of size $|D| - n$) with negative labels only, i.e., $D = \{(a_i, y_i)\}_{i=1}^{|D|} = D^+ \cup D^-$ where $D^+ = \{(a_i, y_i)\}_{i=1}^{n}$ and $D^- = \{(a_i, y_i)\}_{i=n+1}^{|D|}$.

For convenience, we use the following short-hand notation:

$$v' = \log(\frac{1}{\epsilon + s} + 1), \quad v'' = \log(\frac{1}{\epsilon} + 1), \quad y' = \log(\epsilon + s) - \frac{v'}{2}, \quad y'' = \log(\epsilon) - \frac{v''}{2}.$$

Observing a positive example is equivalent to observing $y'$ with noise $\mathcal{N}(0, v')$ on $f_\alpha$ and observing $y''$ with noise $\mathcal{N}(0, v'')$ on $f_\beta$. Vice versa for observing a negative example. We also denote $1_m$ as a column vector of size $m$ filled with 1s, and $I_m$ as an identity matrix of size $m$.

Recall that $f_\alpha \sim \mathcal{GP}(\mu, k), f_\beta \sim \mathcal{GP}(\mu, k)$ and $f_\alpha \perp\!\!\!\perp f_\beta$. The posterior for $f_\alpha$ is $f_\alpha \mid D \sim \mathcal{GP}(\mu_\alpha, k_\alpha)$. For any $a, a' \in \mathcal{A}$,

$$\mu_\alpha(a) = \mu(a) + k(a, \boldsymbol{a})K_\alpha^{-1}(\boldsymbol{y}_\alpha - \mu(\boldsymbol{a})), \quad k_\alpha(a, a') = k(a, a') - k(a, \boldsymbol{a})K_\alpha^{-1}k(\boldsymbol{a}, a'), \quad (5)$$

where

$$k(a, \boldsymbol{a}) = [k(a_i, a)]_{i=1}^{|D|} \in \mathbb{R}^{1 \times |D|}, \quad k(\boldsymbol{a}, a') = [k(a_i, a')]_{i=1}^{|D|} \in \mathbb{R}^{|D| \times 1},$$

$$\mu(\boldsymbol{a}) = [\mu(a_i)]_{i=1}^{|D|} \in \mathbb{R}^{|D| \times 1}, \quad K = [k(a_i, a_j)]_{i=1,j=1}^{|D|} \in \mathbb{R}^{|D| \times |D|},$$

and

$$\boldsymbol{y}_\alpha = \begin{bmatrix} y'1_n \\ y''1_{|D|-n} \end{bmatrix} \in \mathbb{R}^{|D| \times 1}, \quad K_\alpha = K + \begin{bmatrix} v'I_n & 0 \\ 0 & v''I_{|D|-n} \end{bmatrix} \in \mathbb{R}^{|D| \times |D|}.$$

Similarly for $f_\beta \mid D \sim \mathcal{GP}(\mu_\beta, k_\beta)$, we have

$$\mu_\beta(a) = \mu(a) + k(a, \boldsymbol{a})K_\beta^{-1}(\boldsymbol{y}_\beta - \mu(\boldsymbol{a})), \quad k_\beta(a, a') = k(a, a') - k(a, \boldsymbol{a})K_\beta^{-1}k(\boldsymbol{a}, a'), \quad (6)$$

where

$$\boldsymbol{y}_\beta = \begin{bmatrix} y''1_n \\ y'1_{|D|-n} \end{bmatrix} \in \mathbb{R}^{|D| \times 1}, \quad K_\beta = K + \begin{bmatrix} v''I_n & 0 \\ 0 & v'I_{|D|-n} \end{bmatrix} \in \mathbb{R}^{|D| \times |D|}.$$

Since $f = f_\alpha - f_\beta$, by combining Eq. 5 and Eq. 6, we have $f \mid D \sim \mathcal{GP}(\mu_D, k_D)$, and

$$\mu_D(a) = \mu_\alpha(a) - \mu_\beta(a) = k(a, \boldsymbol{a})\left(K_\alpha^{-1}(\boldsymbol{y}_\alpha - \mu(\boldsymbol{a})) - K_\beta^{-1}(\boldsymbol{y}_\beta - \mu(\boldsymbol{a}))\right),$$

$$k_D(a, a') = k_\alpha(a, a') + k_\beta(a, a') = 2k(a, a') - k(a, \boldsymbol{a})\left(K_\alpha^{-1} + K_\beta^{-1}\right)k(\boldsymbol{a}, a'). \quad (7)$$

Thus, we obtain the closed-form posterior for function $f$.

For classifier $g(a) = \frac{1}{1+e^{-f(a)}}$, we can then get its PDF as follows,

$$p_{g(a)}(y) = \frac{1}{y(1-y)\sqrt{2\pi k_D(a, a)}} \exp\left(-\frac{(\log(y) - \log(1-y) - \mu_D(a))^2}{2k_D(a, a)}\right). \quad (8)$$

### B.2 Posterior inference for weights (extension of §3.3.3)

If we use the cosine kernel in §3.3.3, the posterior of $f_\alpha$ can be written as

$$f_\alpha(a) = W_\alpha^\top \psi(a) + \mu(a), \text{ where } W_\alpha \mid D \sim \mathcal{N}(u_\alpha, \Sigma_\alpha), W_\alpha \in \mathbb{R}^{d+1}.$$

This means $f_\alpha(a) \mid D \sim \mathcal{N}(u_\alpha^\top \psi(a) + \mu(a), \psi(a)^\top \Sigma_\alpha \psi(a))$.

Because of Eq. 5 and Eq. 4, we can also write the posterior of $f_\alpha$ as

$$\mu_\alpha(a) = \mu(a) + \psi(a)^\top \psi(\boldsymbol{a}) K_\alpha^{-1}(\boldsymbol{y}_\alpha - \mu(\boldsymbol{a})),$$
$$k_\alpha(a, a') = \psi(a)^\top \psi(a) - \psi(a)^\top \psi(\boldsymbol{a}) K_\alpha^{-1} \psi(\boldsymbol{a})^\top \psi(a).$$

By comparing the above two ways of writing the posterior of $f_\alpha$, we obtain

$$u_\alpha = \psi(\boldsymbol{a}) K_\alpha^{-1}(\boldsymbol{y}_\alpha - \mu(\boldsymbol{a})), \quad \Sigma_\alpha = I_{d+1} - \psi(\boldsymbol{a}) K_\alpha^{-1} \psi(\boldsymbol{a})^\top.$$

Similarly, for $f_\beta(a) = W_\beta^\top \psi(a) + \mu(a), W_\beta \mid D \sim \mathcal{N}(u_\beta, \Sigma_\beta)$, we have

$$u_\beta = \psi(\boldsymbol{a}) K_\beta^{-1}(\boldsymbol{y}_\beta - \mu(\boldsymbol{a})), \quad \Sigma_\beta = I_{d+1} - \psi(\boldsymbol{a}) K_\beta^{-1} \psi(\boldsymbol{a})^\top.$$

Then, for $f = f_\alpha - f_\beta = W^\top \psi(a), W \mid D \sim \mathcal{N}(\mu, \Sigma)$, we have

$$u = \psi(\boldsymbol{a}) \left( K_\alpha^{-1}(\boldsymbol{y}_\alpha - \mu(\boldsymbol{a})) - K_\beta^{-1}(\boldsymbol{y}_\beta - \mu(\boldsymbol{a})) \right), \Sigma = 2I_{d+1} - \psi(\boldsymbol{a}) \left( K_\alpha^{-1} + K_\beta^{-1} \right) \psi(\boldsymbol{a})^\top.$$

This means we can directly sample classifiers from a Beta GP with a cosine kernel by sampling weights $W$ from a multivariate Gaussian distribution defined above.

### B.3 How to set hyperparameters

There are two hyperparameters in GPP with the cosine kernel: $\epsilon$, which determines the prior, and $s$, which determines the posterior.

For any $a \in \mathcal{A}$, the prior on the probability that the label is positive is $\text{Beta}(\epsilon, \epsilon)$. As noted in §3.3.1, $\epsilon < 1$ reflects a belief that $g(a)$ should be close to either 0 or 1; $\epsilon = 1$ gives a uniform distribution over $[0, 1]$; and $\epsilon > 1$ reflects a belief that $g(a)$ is centered at 0.5. In the Beta GP, the Beta prior is approximated as

$$p_{g(a)}(y) = \frac{1}{y(1-y)\sqrt{4\pi \log(\frac{1}{\hat{\epsilon}} + 1)}} \exp \left( -\frac{(\log(y) - \log(1-y))^2}{4 \log(\frac{1}{\hat{\epsilon}} + 1)} \right). \tag{9}$$

Eq. 9 is obtained using the prior of $f$, i.e., $\mu_D(a) = 0, k_D(a, a) = 2k(a, a) = 2\log(\frac{1}{\hat{\epsilon}} + 1)$, in Eq. 8. Users can choose $\text{Beta}(\hat{\epsilon}, \hat{\epsilon})$ for moment matching in Eq. 9 to get a better approximation of $\text{Beta}(\epsilon, \epsilon)$.

Figure 9 shows both the PDF of Beta priors and the approximations.

For setting the hyperparameter $s$, users can also directly inspect the behaviors of different values of $s$ and choose an appropriate value. Figure 10 and Figure 11 show how the posterior changes with one negative or two opposite-label observations. Larger $s$ leads to more concentrated posterior.

Since all of these distributions are easily computable and can be visualized clearly, users can directly inspect the behaviors of these different hyperparameters and choose a suitable option.

### B.4 Analyses of episteme (extension of §3.4)

For each $a \in \mathcal{A}$, episteme is the negative of $\mathbb{H}[g(a)]$. By Eq. 8, we have

$$\begin{aligned}
\mathbb{H}[g(a)] &= -\int p_{g(a)}(y) \log p_{g(a)}(y) \, \mathrm{d}y \\
&= -\mathbb{E}[\log p_{g(a)}(y)] \\
&= \mathbb{E}[\log (y(1-y))] + \mathbb{H}[f(a)] \\
&< \mathbb{H}[f(a)].
\end{aligned}$$

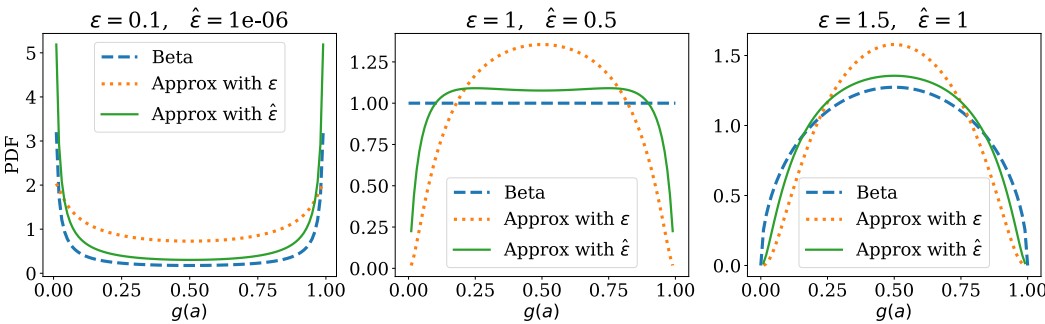

Figure 9: PDF of Beta$(\epsilon, \epsilon)$ and the approximations that either use $\epsilon$ for moment matching or $\hat{\epsilon}$. Because both the PDF of Beta distributions and the approximations in Eq. 9 are easily computable, users can inspect the distributions directly and choose the right $\hat{\epsilon}$ to match with their own beliefs.

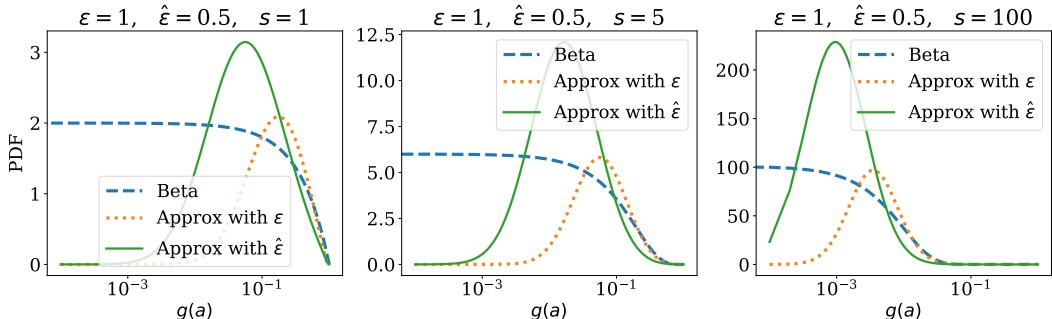

Figure 10: PDF of Beta$(\epsilon, \epsilon + s)$ and the approximates that either uses $\epsilon$ for moment matching or $\hat{\epsilon}$. These distributions are the (approximated) posteriors of $g(a)$ after observing 1 negative example. Hyperparameter $s$ are 1 (Left), 5 (Middle) or 100 (Right), and with a larger $s$, the approximate becomes more concentrated at a lower value of $g(a)$.

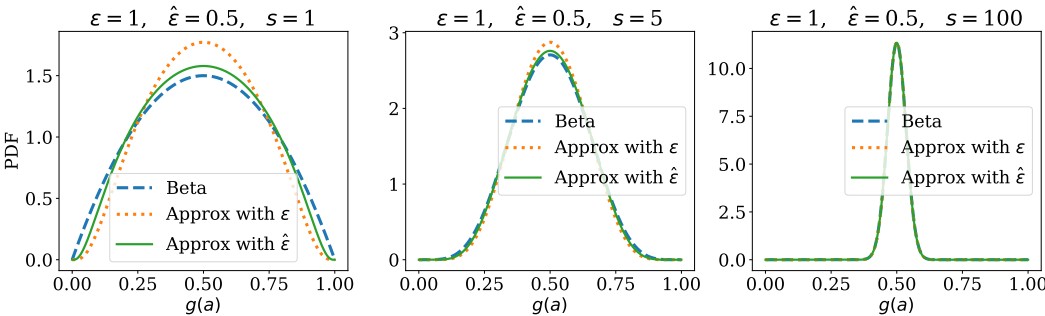

Figure 11: The same setup as Figure 10, except that the observations include 1 positive and 1 negative examples at the same representation $a$. A larger $s$ results in a PDF that is more concentrated at $g(a) = 0.5$.

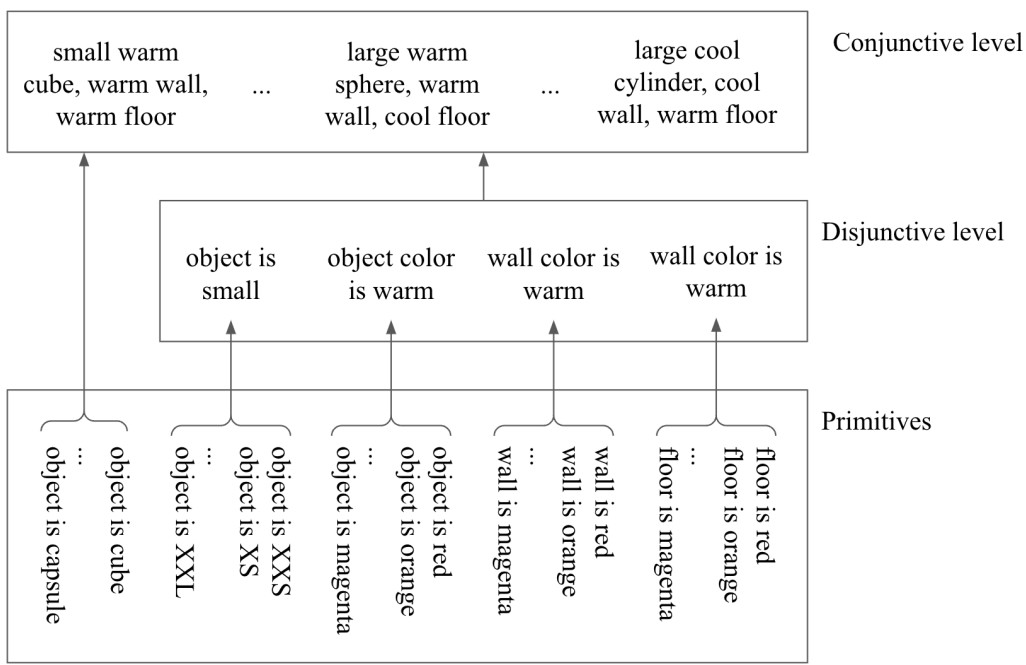

Figure 12: Ontology of training labels for the 3D Shapes dataset [Burgess and Kim, 2018].

The last inequality is because $y(1-y) < 1$, i.e., $\log(y(1-y)) < 0$.

In §3.3.1, we set a constraint on the kernel $k$ such that $k(a,a) = \log(\frac{1}{\epsilon} + 1)$. By Eq. 7, the entropy of $f(a)$ can be bounded as follows.

$$\mathbb{H}[f(a)] = \frac{1}{2}\log(2\pi e k_D(a,a)) \leq \frac{1}{2}\log(4\pi e \log(\frac{1}{\epsilon} + 1)).$$

Hence there exists a lower bound on episteme for GPP. However, $\mathbb{H}[g(a)]$ can approach $-\infty$ because (1) variable $y$ can be infinitely close to 0 or 1, and (2) $k_D(a,a)$ can also be infinitely close to 0, which means episteme has no finite upper bound.

In natural language, our analyses of episteme show that ignorance has a limit, but knowledge has no limit. This is a widely recognized idea, and it is also reflected in the words of Zhuangzi, a Chinese philosopher from the 4th century BCE: "Your life has a limit, but knowledge has none."

## C Experiment details

In this section, we include details on experiment setups and additional results.

### C.1 3D Shapes ontology

The ontology of training labels for the 3D Shapes dataset [Burgess and Kim, 2018] is illustrated in Figure 12. Images are generated from 6 ground truth independent primitives: 10 floor colors, 10 wall colors, 10 object colors, 8 scales, 4 shapes and 15 orientations of the shapes (orientation is excluded from the ontology since it's only distinguishable for cubes). The disjunctive level of the ontology groups together ranges of color and shape primitives into binary concepts: warm/cool and small/large, respectively. Concepts in the conjunctive level are the Cartesian product of concepts in the disjunctive level and the shape primitives.

### C.2 Real-world OOD detection

In-distribution (ID) queries are sampled disjointly from the validation split of the ImageNet dataset [Russakovsky et al., 2015], where the probe observes 10 sets of $D$s using 10 binary classification

tasks defined by ImageNet superclasses. These superclasses are defined by building a tree using the WordNet hierarchy [Miller, 1994] where the leaves of this tree are ImageNet classes (e.g., the superclass "dog" contains Chihuahua, Japanese Spaniel, Maltese, etc.). The ten classification tasks we use are: (1) dog vs snake, (2) fish vs lizard, (3) bird vs snake, (4) dog vs bird, (5) cat vs bird, (6) fish vs snake, (7) bird vs fish, (8) snake vs lizard, (9) cat vs dog, (10) bird vs lizard.

Out-of-distribution (OOD) images are generated with pixel-wise uniform random noise. This noise is passed through the basis function $\phi$ to construct the OOD query.

### C.3  Relations between judged probability, episteme and alea

In this section, we present more empirical results. The experiment setting is the same as §4.3.

First, we evaluate how judged probability and alea change as episteme increases, and how judged probability changes as alea increases. Figure 13 and Figure 14 show the results for GPP and LPE respectively. Please use a higher quality PDF viewer if the points don't show up in the figures. Each row corresponds to a different ground truth probability, which means the probability that an originally positive stimulus remains to have positive labels in observations, when we manually inject fuzziness to concepts in the 3D Shapes dataset [Burgess and Kim, 2018]. So in the ideal case, judged probability predictions should converge to the ground truth probability for stimuli that are originally positive. Each scattered point corresponds to the predictions on a stimulus that is originally positive.

GPP consistently produces rational uncertainty measures. There are no extreme judged probability predictions with low episteme. Alea converges to low values for 1.0 ground truth probability and higher values when the ground truth probability is 0.25 or 0.75, and alea converges to the highest values when the ground truth probability is 0.5. These are all expected since with 0.5 ground truth probability, the level fuzziness reaches the highest.

On the contrary, LPE tends to have more extreme predictions on judged probability no matter what the ground truth probability is. However, the average judged probability of LPE does get affected by the ground truth probability. For example, when the ground truth probability is 0.25, more masses of judged probability accumulate between 0 to 0.2. This means the average judged probability can be close to 0.25. Similarly, when the ground truth probability is 0.5, about half of the predictions of judged probability are between 0.8 to 1.0, and the other half are between 0.0 to 0.2. While this ensures the judged probability is close to ground truth probability if we average over all stimuli, the individual predictions cannot be used to evaluate the fuzziness of concepts.

Figure 15 shows AUROC, AUPRC and accuracy of GPP and LPE for different numbers of observations. These metrics are averaged over both positive and negative queries. Interestingly, even when the ground truth probability is 0.25 (only 1/4 of the positive examples remain positive), GPP can still achieve almost 1.0 AUROC and AUPRC. As expected, the accuracy of GPP is about 0.5 for 0.25 ground truth probability (since the judged probability predictions are mostly lower than 0.5 for the positive examples, and the negative examples are almost all correct). But because LPE does not always have low judged probability predictions even if ground truth probability is 0.25, its accuracy is higher than 0.5.

These results confirm the rationality and good performance of GPP as a valid probing method.

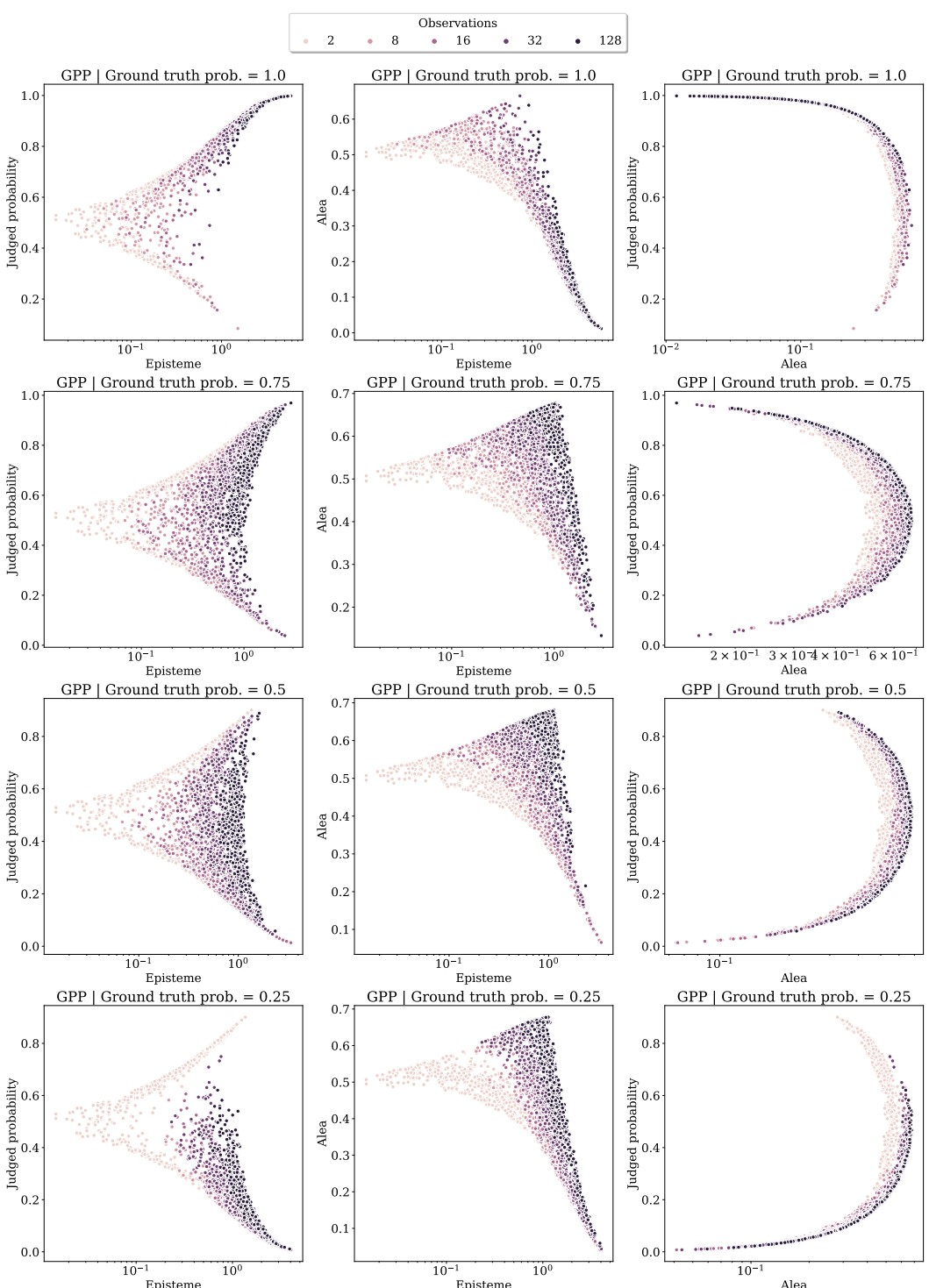

Figure 13: Relationships between judged probability, episteme and alea using GPP.

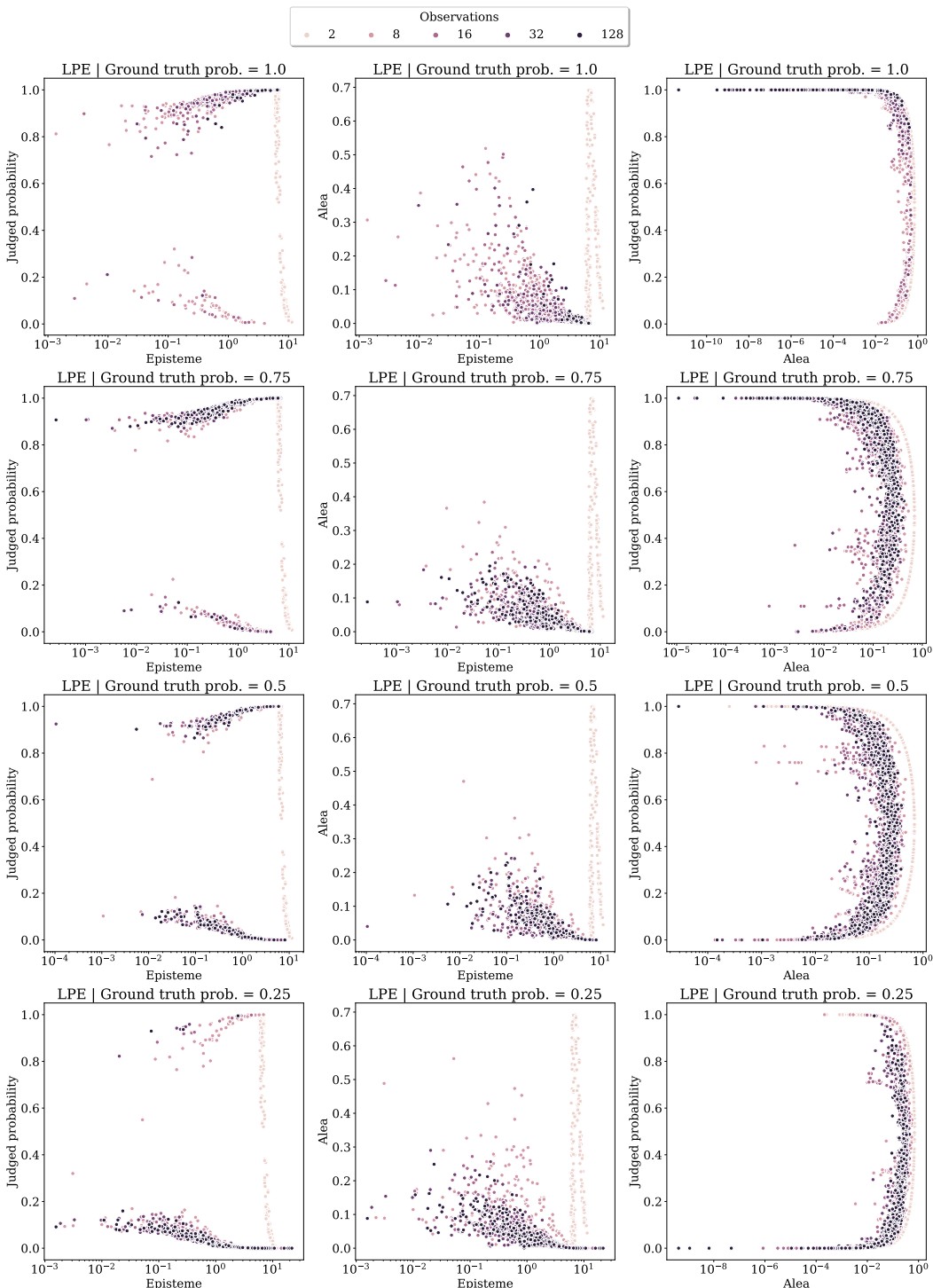

Figure 14: Relationships between judged probability, episteme and alea using LPE.

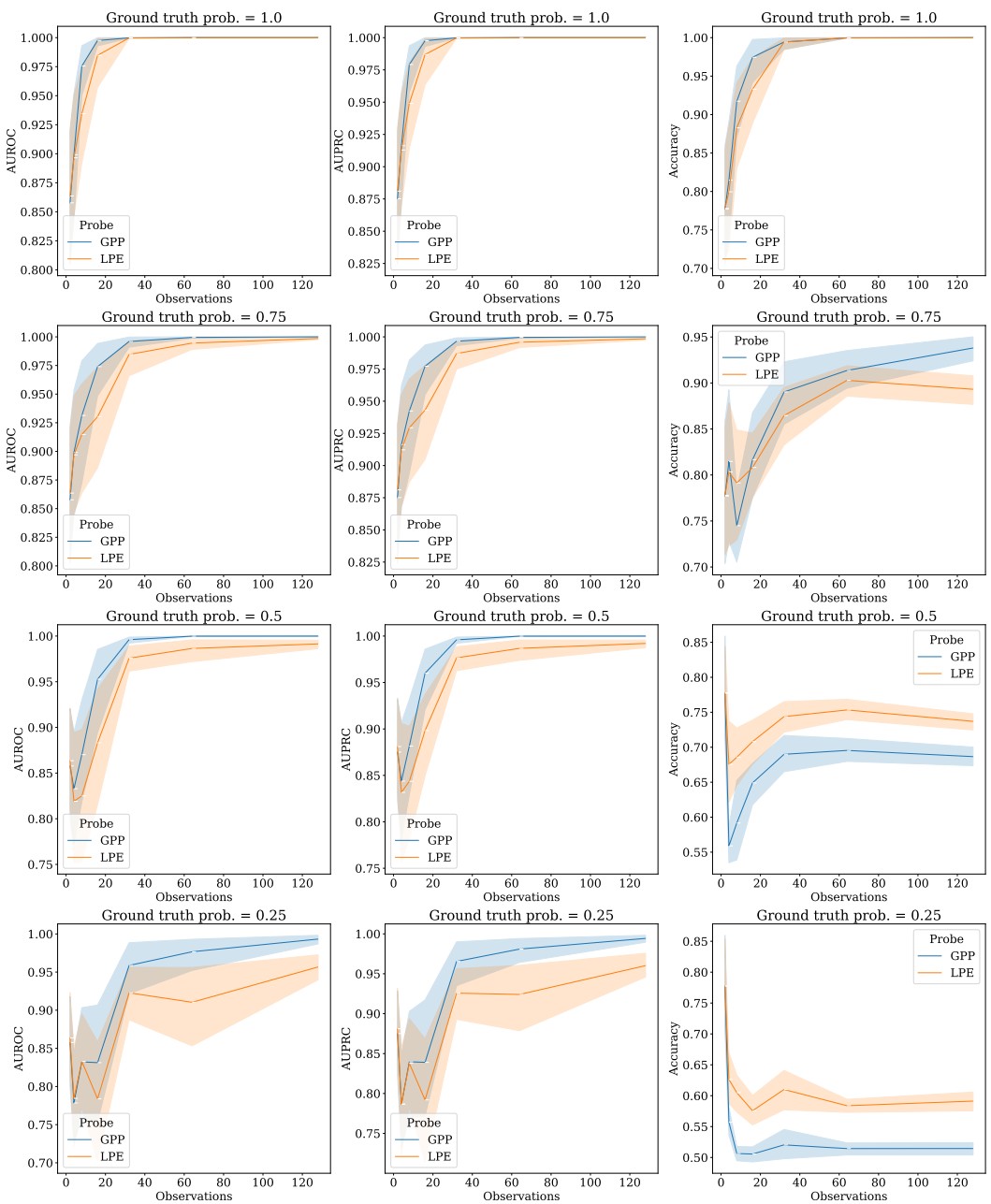

Figure 15: AUROC, AUPRC and accuracy of GPP and LPE using different numbers of observations.

