# OpenReview forum: "Gaussian Process Probes (GPP) for Uncertainty-Aware Probing"
_NeurIPS.cc/2023/Conference — NeurIPS 2023 poster_

### Official Review · Reviewer_zzXk · 2023-06-19

**Soundness:** 3 good
**Presentation:** 4 excellent
**Contribution:** 3 good
**Rating:** 7
**Confidence:** 3

**Summary:**

This paper provides Gaussian process probes (GGP), a probabilistic method to evaluate uncertainty for a binary classification task over a (pre-trained) feature extractor. The core idea is to use GP instead of a linear probe on the feature extractor. The use of GP provides a natural way to estimate two uncertainty measures: aleatory and epistemic. To examine the performance of GPP, the authors conduct two experiments using synthetic and real images, showing that GPP can preferably estimate uncertainty.

**Strengths:**

1. Novel formulation to evaluate uncertainty for linear probe problems.
2. The paper is well-written and easy to follow. A lot of illustrative figures help to understand the key concepts.
3. Solid derivation of the GP model and uncertainty estimation.

**Weaknesses:**

1. The experiment results (Fig 5) look somewhat strange. For me, the LPE result doesn't make sense. The setting here is that only positive labels are noisy (flipped to negative with 50% probability), and the negative labels do not contain noise. In such a situation, the well-trained classifier will output 0.5 as the judged probability for positive and 0 for negative examples. This should occur when the classification problem is linearly separable on the feature space, and the number of observations is large enough. I assume both conditions are satisfied when the number of observations is 128, but I cannot see such a tendency from LPE. Also, due to the imbalanced noise, I expect to see a vertically asymmetric behavior for LPE (e.g. the upper limit of judged probability would be close to 0.5), but it is not.
2. This paper doesn't explicitly explain multiclass cases. Line 111 says extending GPP for multiclass problems is straightforward, but I don't feel it is trivial. Also there are no experiments for multiclass cases.
3. Some technical descriptions are unclear. See "Questions" below.

**Questions:**

1. In 154, it is said the kernel k is defined as "k(a, a) = v", but the definition (3) tells us that  "k(a, a) = v (||a||^2 + 1) / (||a|| + 1)^2", which is not v unless a = 0. Why does this inconsistency happen?
2. Line 144 says that GP with the kernel (3) is "equivalent to defining a distribution over linear latent functions", but then why does Fig 1 show some nonlinearity?
3. What is the training procedure of LPE in Fig 5? Did you confirm that LPE was properly trained?

**Limitations:**

The authors fairly address the limitations.

---

> ### Author Rebuttal · Authors · 2023-08-09
>
> We thank the reviewer for the positive feedback.
>
>
> > LPE results in Figure 5, “the well-trained classifier will output 0.5 as the judged probability for positive and 0 for negative examples. This should occur when the classification problem is linearly separable on the feature space, and the number of observations is large enough. I assume both conditions are satisfied when the number of observations is 128, but I cannot see such a tendency from LPE.”
>
>
> In our experiments, we randomly flipped the labels for the positive class, but it is possible for the linear classifier to find features that linearly separate the activations with new labels. This is especially true if the dimension of the activations is larger than the number of observations. For example, a hyperplane in 3D almost always exists to separate 2 points. Since the number of dimensions for the activations is 256, and the number of observations is 128, it is very likely that there exists a hyperplane that separates the activations (unless the activations with positive labels are almost overlapping). Also note that points in high dimensions are naturally distant from each other. Roughly speaking, for logistic regression, it only requires the distance to be larger than 3 to predict the judged probability to be larger than 95% or lower than 5%. So it is not surprising that judged probability predictions of LPE are close to 0 or 1.
>
>
>
>
> > “due to the imbalanced noise, I expect to see a vertically asymmetric behavior for LPE (e.g. the upper limit of judged probability would be close to 0.5), but it is not.”
>
>
> As mentioned above, it is very likely for LPE to find a hyperplane that separates activations with the new labels. And since the queries are sampled from the same distribution as the observations with original labels, LPE is still going to output probabilities close to 0 or 1.
>
>
>
>
> > “ Line 111 says extending GPP for multiclass problems is straightforward, but I don't feel it is trivial. Also there are no experiments for multiclass cases”
>
>
> Extending the Beta GP component of GPP to multiclass only requires switching the Beta distribution to the Dirichlet distribution (Beta distribution is a special case of Dirichlet), and we can achieve this by using k latent functions instead of 2, where k is the number of classes. That is what we meant by being straightforward, and we will make this clearer in the paper.
>
>
> We aim to detect whether a model is able to represent a concept or not (probing as binary classification), so multiclass classification was not the best problem setup for us. However, we agree that the multiclass problem is a very important topic, and we need to formulate the problem carefully and spend more effort to understand and build tools for multiclass probing (in addition to the GP classification component). For example, some problems may require multiclass multilabel outputs, where the binary version of GPP can be directly plugged in for each label. But for multiclass single-label problems, we might need to redefine episteme for each label and understand the new relationship of episteme, alea and judged probability. We hope to explore the multiclass probing problem more in our future work, and we will add discussions about these considerations in the next version of this paper.
>
>
>
>
> > “1. In 154, it is said the kernel k is defined as "k(a, a) = v", but the definition (3) tells us that "k(a, a) = v (||a||^2 + 1) / (||a|| + 1)^2", which is not v unless a = 0. Why does this inconsistency happen?”
>
>
> Thank you for pointing this out. We made a typo in Equation (3). There should be a square root in the denominator to properly normalize, i.e., $k(a, a') = v\frac{a^T a' + 1}{(\lVert a\rVert^2+1)^{\frac12}(\lVert a'\rVert^2+1)^{\frac12}}$, which ensures that $k(a,a)=v$. This was implemented properly in our code (line 191 of code/gp.py, which is used by line 211 in the cosine_kernel function). We will correct this in the paper.
>
>
>
>
> > “2. Line 144 says that GP with the kernel (3) is "equivalent to defining a distribution over linear latent functions", but then why does Fig 1 show some nonlinearity?”
>
>
> This is because the linear latent functions ($f_\alpha$ in Equation 4) are only linear to the *normalized activations with bias terms*. The latent functions are nonlinear in the space of activations.
>
>
>
>
> > “3. What is the training procedure of LPE in Fig 5? Did you confirm that LPE was properly trained?”
>
>
> LPE is a bootstrap ensemble of linear probes. For each member of the ensemble, we trained a logistic regression classifier on a dataset sampled from the original set of observations with replacement. The size of the dataset was the same as the number of observations. Each ensemble has 100 members. We also made sure to include at least one positive and one negative example so that logistic regression is possible. We will make these clear in the new version of the paper.
>
>
> To validate the correctness of our LPE implementation, we performed sanity checks using LPE for model M.1 and task P.1 with no fuzzy labels. In this sanity check, LPE was able to achieve AUROC=1 with 40 or more observations. We include this result in the PDF attached to the “global response”, and the full figure can be found in Figure 15 in the appendix. Figure 14 of the appendix also illustrates that judged probabilities predicted by LPE are close to 1 for the majority of positive examples when ground truth probability is 1; if the ground truth probability is 0.25, the judged probabilities predicted by LPE are closer to 0. Collectively, these pieces of evidence support that LPE was properly trained in our experiments.

---

> > ### Comment · Reviewer_zzXk · 2023-08-12
> > **Raise my score**
> >
> > Thank you to the authors for the response. The explanations and the additional experiments about LPE make sense. My major concerns are resolved, and I will raise my score.

---

### Official Review · Reviewer_VzJz · 2023-07-04

**Soundness:** 3 good
**Presentation:** 3 good
**Contribution:** 3 good
**Rating:** 6
**Confidence:** 2

**Summary:**

This work introduces a unified framework called Gaussian process probes (GPP) for probing and quantifying uncertainty in models' representations of concepts. GPP extends linear probing methods and uses a Bayesian approach to estimate the distribution of classifiers induced by the model. This distribution measures the model's ability to represent concepts and provides a measure of confidence in the representation. GPP is a simple procedure applicable to any pre-trained models with vector representations, requiring no access to training data, gradients, or architecture details. The validation experiments on synthetic and real-world datasets demonstrate that GPP can effectively probe concept representations with a small number of examples, accurately measure both epistemic (confidence) and aleatory (fuzziness) uncertainties, and detect out-of-distribution data.

**Strengths:**

The paper addresses an important problem of understanding the inner representations of complex models. It does so by providing some estimates over two different types of uncertainty. It gives clear explanations for the epistemic and aleatoric uncertainties. The use of GPs and the study of two types of uncertainty in this particular context seems novel.



**Weaknesses:**

As a non-expert in the area of probing, I found some parts of the paper hard to follow (see Questions below).

**Questions:**

Can you explain in more detail the choice of prior as outlined in Section 2.3.1. I'm not sure I understand what you mean by matching the Beta prior and matching the normal distribution in lines 125-126. I think this must impose important assumptions on the model, affecting the outcomes so it would be good to communicate it clearly.

Are there any practical implications to considering the two types of uncertainty separately? Are there any cases where it might give counterintuitive results?

Do you foresee any practical issues with using higher dimensional Dirichlet-based GPs? Can the inference still be performed in closed form?

**Limitations:**

The authors outlined limitations in the main part of the paper and provided a broader impact statement. From a practitioner's point of view, it would be important to outline the exact assumptions that the GP model imposes (I assume those will depend on the specific dataset/model). I believe this is done to some extent in the paper but I have further questions (as mentioned above).

The quality of the writing could be improved, it requires a proof-read.

Minor:
Line 69 - best repeated
Line 124 - uniformly distributed

---

> ### Author Rebuttal · Authors · 2023-08-09
>
> We thank the reviewer for the positive feedback.
>
>
> > “Can you explain in more detail the choice of prior as outlined in Section 2.3.1. I'm not sure I understand what you mean by matching the Beta prior and matching the normal distribution in lines 125-126.”
>
>
> The goal is to use a Log-normal distribution to approximate a Gamma distribution. Because a Beta variable can be written as two independent Gamma variables, we can approximate the Beta variable with two Log-normal variables. The logarithms of Log-normal variables are distributed according to normal distributions. That is why we can approximate a Beta distribution with two normal distributions. Please see more details on how close the PDF approximations are in Section B.3 (especially Figure 9) in the appendix included in the supplementary material.
>
>
>
>
> > “Are there any practical implications to considering the two types of uncertainty separately? Are there any cases where it might give counterintuitive results?”
>
>
> The practical implications are that we can now distinguish between fuzziness in concepts (i.e., aleatory) and not having enough information to reveal what the concept is (i.e., epistemic). In the past literature, if a linear probe predicts 0.5 probability for the binary case, we would conclude that the model does not represent the concept. However, in this work, we showed that predicting 0.5 does not mean that the model cannot represent the concept, since some concepts can be intrinsically fuzzy. For example, some people think tomatoes are fruits but others don’t, but that does not mean people don’t have representations for fruits. Another situation is that the probe does not have enough observations to tell what the concept is. Please see the figure and 2nd paragraph in the introduction for more insights. In short, the predictions from GPP can reveal much richer information than previous methods, and the rich information provides practical insights on what the model is able to represent. We have not seen cases where GPP gives counterintuitive results.
>
>
>
>
> > “Do you foresee any practical issues with using higher dimensional Dirichlet-based GPs? Can the inference still be performed in closed form?”
>
>
> For multiclass problems using Dirichlet-based GPs, the number of latent functions is the same as the number of classes. Hence the computation for inference grows linearly with the number of classes. The inference can still be performed in closed form, since those latent functions are independently distributed according to Gaussian processes, and the inference for each GP can be done in closed form. For higher dimensional activations, we don’t expect any issues since the GP computations rely on the kernel and the kernel only requires computing norms or inner products of activations.
>
>
>
>
> > “writing” and typos
>
>
> Thanks for pointing this out. We will further polish the paper with more rounds of proof-read.

---

> > ### Comment · Reviewer_VzJz · 2023-08-18
> >
> > Thank you for the clarifications. I believe the paper raises interesting points and shows some appealing results. However, I am not able to evaluate the significance of the contribution in the context of existing literature, hence my low confidence stands.

---

### Official Review · Reviewer_3D6e · 2023-07-09

**Soundness:** 3 good
**Presentation:** 2 fair
**Contribution:** 2 fair
**Rating:** 5
**Confidence:** 2

**Summary:**

The manuscript proposes a Gaussian process-based probing (monitoring a layer using only the layer’s activations without influencing the model itself) method that can estimate uncertainty in prediction. In the experimental result section, the proposed method is applied to several example datasets that can check whether the proposed method can correctly estimate fuzziness in concepts and to out of distribution (OOD) detection.

**Strengths:**

I think that the text is easy to read and understand. This work can be significant because we need more tools to understand deep neural networks, which are black-box models. Uncertainty quantification is also a major selling point of the proposed method. However, I hope that the manuscript would have provided monitoring and diagnostic tools to understand layers’ behaviors using estimated uncertainty.

**Weaknesses:**

1. I think that the novelty of the proposed method is incremental. In fact, a number of works have been proposed based on GP or Deep GP models for uncertainty estimation (including decomposing uncertainty into Aleatoric and Epistemic uncertainty) and out-of-distribution (OOD) detection. The authors put together several ideas (from existing work) to create a new method that can be a good tool for certain problems. However, I did not find any new innovations or improvements from the methodologies of the proposed method.

2. I think that the manuscript could be further improved in terms of presentation. My main concern is that the motivation for the use of Beta (Dirichlet) Gaussian processes is not clearly stated in the main text. Why not just use the original Gaussian process formulation for classification? This approach requires approximation for inference, but there already have been accurate and practical tools, e.g., expectation and propagation (EP). If the main motivation is about computational complexities (as in the reference paper for Beta GP [ref Milios et al., 2018]), then the examples included in the experiments do not seem to be well designed to get the readers appreciate this motivation from them (as the numbers of the training datasets appear to be small).


**Questions:**

In eq. (4), what is \mu(a)? I could not find its definition in the text.

Regarding eq. (4) (and eq. (2)), it appears that we don’t need to consider two linear functions, W_alpha and W_beta, because they can be reduced a single linear function f = f_alpha - f_beta = W^T \psi(x), where W=W_alpha - W_beta. What did I miss here?


**Limitations:**

Yes. No potential negative societal impact of their work.

---

> ### Author Rebuttal · Authors · 2023-08-09
>
> We thank the reviewer for the review and constructive feedback.
>
>
> We would like to emphasize the novelty of this work comes from the application of kinds of uncertainty and GPs in the context of probing. . Our goal is not simply to measure the uncertainty of predictions from a neural net, where we agree GPs have previously been used. Rather, we designed GPP (Gaussian process probes) to measure uncertainty related to  any "concepts" represented by a neural net from its activations -- a new way to "probe" neural nets. To the best of our knowledge, delineating different kinds of uncertainty for deep neural net representations via probing has not been studied in the past. Both intrinsic fuzziness in concepts and uncertainty as to which concept applies are ubiquitous cognitive phenomena, making it natural to ask how these quantities can be extracted from neural nets. When learning new concepts, it is natural for people to perform OOD detection (i.e., more colloquially, being able to say “I’m not sure since I haven’t learned it yet”). GPP fills in the gap between the existing probing literature, which has measured the presence or absence of concepts but not uncertainty about those concepts, and people’s intuitive representation of concepts (where uncertainty of different kinds can easily be articulated). This is an important step for advancing the capabilities of explainable and interpretable AI.
>
>
>
>
> > “monitoring and diagnostic tools to understand layers’ behaviors using estimated uncertainty”
>
>
> We agree with the reviewer that deeper analyses and comparisons between layers are exciting directions for follow-on work. Since GPP can be used for any layers of activations in a deep neural network, our work builds a strong foundation from which to  pursue these directions in the future.
>
>
>
>
> > “novelty” from a GP perspective
>
>
> It is straightforward to distinguish the two kinds of uncertainty in the **GP regression** framework. However, to the best of our knowledge, aleatory uncertainty (fuzziness of concepts) has not been studied in the **GP classification** literature. There also has not been previous work in the GP literature to establish the correspondence between human perceptions of uncertainty [Fox and Ülkümen,2011] and GP classification predictions.
>
>
>
>
> > “the motivation for the use of Beta (Dirichlet) Gaussian processes”, “Why not just use the original Gaussian process formulation for classification?”
>
>
> Thank you  for raising this point. Beta (or Dirichlet) Gaussian process classifiers have been shown to either outperform or achieve similar performance than classic GP classification approximations [Milios et al., 2018]. While our work is not limited to Beta GPs, we used the Beta (Dirichlet) GPs mainly because to the best of our knowledge, the Beta (Dirichlet) GPs is the state-of-the-art GP classification method. The other reason we decided to use the Beta GP is because the GP posterior inference can be done in closed form, and we can easily inspect the observations and posteriors to make sure the Beta GP is doing what we intended to do. We will make this motivation clearer in the paper.
>
>
>
>
> > “In eq. (4), what is \mu(a)?”
>
>
> $\mu$ is the mean function of the Beta GP first mentioned in the last paragraph of Section 2.2, and defined in Section 2.3.1. We will make this clearer for Eq. (4).
>
>
>
>
> > “Regarding eq. (4) (and eq. (2)), it appears that we don’t need to consider two linear functions, W_alpha and W_beta, because they can be reduced a single linear function f = f_alpha - f_beta = W^T \psi(x), where W=W_alpha - W_beta. What did I miss here?”
>
>
> That’s a very nice observation. We simplified the expression to a single linear function, but we have to use two different functions in the Beta GP, because each observed datapoint  transforms to two pseudo datapoints with heteroscedastic noises, one for $f_\alpha$ and the other for $f_\beta$ (Section 2.3.2). During inference, the posteriors of $f_\alpha$ and $f_\beta$ are updated accordingly. If we only use one function $f$, we cannot directly obtain the posterior of $f$ without updating $f_\alpha$ and $f_\beta$. Please see Section B.1 in the appendix for more details.
>
>
>
>
> *References*
>
>
> Craig R Fox and Gülden Ülkümen. Distinguishing two dimensions of uncertainty.Fox, CraigR. and Gülden Ülkümen (2011),“Distinguishing Two Dimensions of Uncertainty,” in Essaysin Judgment and Decision Making, Brun, W., Kirkebøen, G. and Montgomery, H., eds. Oslo:Universitetsforlaget, 2011.
>
>
> Dimitrios Milios, Raffaello Camoriano, Pietro Michiardi, Lorenzo Rosasco, and Maurizio Filippone.Dirichlet-based Gaussian processes for large-scale calibrated classification.Advances in NeuralInformation Processing Systems, 31, 2018.

---

> > ### Comment · Reviewer_3D6e · 2023-08-19
> >
> > I thank to the authors for answering my reviews. I still think that the novelty of the submission is limited from the point of view of mythology. I will stick to my initial rating.

---

> > > ### Author Response · Authors · 2023-08-19
> > >
> > > We appreciate the discussion. In the comment, "mythology" seems to be a typo for methodology. To clarify, our method is based on adapting and applying state-of-the-art methodology in GP classification for **novel applications to solve a new problem** (uncertainty-aware probing) in the area of interpretable and explainable AI. We drew inspirations from cognitive science and probabilistic ML to define, compute and show insights on uncertainty in the context of probing. Novelty is NOT just about methodology. New problem formulations and new applications are also important criteria for novelty. In fact, in the reviewer guidelines (https://nips.cc/Conferences/2023/ReviewerGuidelines), it was pointed out that originality is about new tasks or a novel combination of well-known techniques, besides new methods.
> > >
> > > It would be helpful if the reviewer could point out papers demonstrating that the novelty or contribution (since the score was 2/4) of our work is limited. For example, which papers in the GP classification literature discussed the aleatory uncertainty and its relations to judged probability and epistemic uncertainty, or have shown how to formulate and solve the problem of uncertainty-aware probing?
> > >
> > > Moreover, since the presentation was rated 2/4, it would be great if the reviewer could provide constructive suggestions for improving the presentation. It seems to us that the lack of motivation for Beta GP is not a core issue for "the writing style and clarity, as well as contextualization relative to prior work" in the description of what "Presentation" is in the reviewing guide.

---

### Official Review · Reviewer_iFt6 · 2023-07-16

**Soundness:** 2 fair
**Presentation:** 2 fair
**Contribution:** 2 fair
**Rating:** 6
**Confidence:** 3

**Summary:**

- The authors propose a probabilistic probing method to understand a given pre-trained classifier.
- The authors describe how looking at classifier predicted class probabilities is not enough since "0.5" in a binary task can happen for several reasons spanning the aleatoric/epistemic uncertainty spectrum
- On the other hand, the proposed probabilistic method allows for posterior estimates of classifier's predicted probabilities, which allows for example to produce variance/entropy/etc
- The particular method used is to compute functions g() of a classifier's representation a(x) for inputs x and to study the distribution of g()
- The distribution of g() is defined through a hierarchical GP called the Beta GP
- After giving an exposition of the model the authors introduce two sets of experiments: whether the probe can correctly identify that there is true label uncertainty in some synthetic experiments, and whether the model can correctly identify that data is OOD for a given classifier.

While proposed method is well motivated and properly defined, I have some major questions regarding evaluation (proper definition of all tasks and baselines). For now, I think the paper needs some clarification before acceptance, but would be glad to raise my score, given clarification from authors.

**Strengths:**

The papers strength's are:
- Beta-GP-based probing method is very clearly defined
- All of the computations e.g. posterior computations are precisely stated in the appendix
- The overall motivation of the work (probabilistic probing) is solid
- The introduced entropy/variance based metrics that pull apart some aspects of aleatoric versus epistemic uncertainty, adapted from previous literature, seem like a nice contribution to uncertainty/probing evaluation metrics

**Weaknesses:**

There are two downstream uses of the method
- checking correlation of probes' reported uncertainty versus the true label uncertainty for synthetic/semi-synthetic datasets
- OOD detection

The precise definitions for baselines in uncertainty estimation experiments were not given (namely LPE) and more definitions are necessary to correctly interpret the variance/entropy-based metrics. See "Questions".

For OOD detection, the superiority over baselines is exemplified but the baselines are out of date. The synthetic dataset is not named, and some details are missing from the Imagenet result to understand exactly how to the pretrained model checkpoints were run on the binarized data. See "questions"


Minor writing style suggestion that does not affect my review: In sentences like

"There are important details in Beta GPs that require special attention: how to set the prior and how to approximate the posterior."

Since there are a few distributions floating around that were recently introduced to the reader, it could be helpful to include the symbols in mid-sentence: "There are two important details in Beta GPs that requires special attention: how to set the prior (mu and k(,) in GP(mu,k))
and how to approximate the posterior p(f_alpha,f_beta | D, mu k )."

And include a sentence after

"With samples from the posterior (falpha, fbeta) we can then approximate the distribution of g()~G with the log normals parameterized by f_alpha, f_beta."

This could help the reader piece everything together.

**Questions:**

The results for the first experimental section (uncertainty quantification) seem convincing relative to the baselines but one major issue
- LPE is not cited, and the authors do not mention that they propose it. There is only a high-level description given. I was trying to look for a precise definition of the method since it is taken as one of the main baselines, but cannot find it anywhere.
- Please give a citation for LPE, or precisely define it and state that this is a baseline proposed by the authors.
- This is moreover important since one cannot arbitrarily compare model entropies/variances with each other for continuous models without more assumptions (same base measure, etc). And for this, precise definition of all distributions is important.

For OOD detection, two main groups of concerns, task definition and baselines:

Task issue 1:
    - there seem two be two datasets, a synthetic one and ImageNet
    - this section doesn't explicitly name which synthetic dataset. I only assume by continuity that it is the shapes dataset from the previous subsection. Please name all datasets in all sections/captions/figures they are used in rather than just refer to "synthetic data".
    - The authors say "we generate another set of queries that include 1024 ID images and 1024 OOD images (uniform random
    noise)" but don't say from where is the in-distribution.

Task issue 2:
    - For Imagenet there are also some missing details that stop me from understanding the experiment.
    - I understand this: "Queries and observations are sampled disjointly from the validation split of the ImageNet dataset"
    - I understand this + I read the appendix section: "We make 10 set of Ds using 10 binary classification tasks defined by supersets of ImageNet classes"
    - How are the pre-trained classifiers use? Did you specifically pull checkpoints that were also trained on the binarized problems? Or did you somehow pool together a multiclass classifier's probabilities for the underlying model + renormalize? What happened to the probabilities assigned to the other classes?
    - Please let us know and then revise the paper to clarify the exact use of the pre-trained models versus the data and what exactly was computed, including any equations if there were any transformation steps from multiclass checkpoints to the binary task, or clarify that binary classification model checkpoints were used that correspond to the same binarization that you applied to the data
    - Also, is the out-distribution in both datasets just uniform noise, or uniform noise added to ID data?

Baselines: The baselines seem a little out of date, with the two non-LPE baselines (again, where is LPE from?) being from 2016 and 2018. It's okay to include older baselines as part of a broader evaluation, but why not include also recent methods from 2020-2023 such as any of the below. I might be missing context on this sub-area of ML, but at least in others I review actively, it's fairly rare to find a paper that only evaluates to methods from <=2018. Several well-cited recent methods are documented below. Maybe not all are applicable as baselines for your particular setup/model assumptions/method assumptions (e.g. black box vs having access to something), but please clarify.

    - 2018, ODIN, Enhancing the reliability of out-of-distribution image
    detection in neural networks. https://arxiv.org/abs/1706.02690

    - 2018, A simple unified framework for detecting out-of-distribution samples and adversarial attacks.
    https://papers.nips.cc/paper_files/paper/2018/hash/abdeb6f575ac5c6676b747bca8d09cc2-Abstract.html

    - 2019 Likelihood Ratios for Out-of-Distribution Detection https://proceedings.neurips.cc/paper_files/paper/2019/file/1e79596878b2320cac26dd792a6c51c9-Paper.pdf

    - 2020, Energy-based out-of-distribution detection. https://proceedings.neurips.cc/paper/2020/hash/f5496252609c43eb8a3d147ab9b9c006-Abstract.html

    - 2021, REACT, React: Out-of-distribution detection with rectified activations.
    https://proceedings.neurips.cc/paper/2021/hash/01894d6f048493d2cacde3c579c315a3-Abstract.html

    - 2022, Dice: Leveraging sparsification for out-of-distribution detection. In ECCV,
    https://arxiv.org/abs/2111.09805

    - 2022, Out-of-distribution detection with deep nearest neighbors. https://arxiv.org/abs/2204.06507

    - 2022, Scaling out-of-distribution detection for real-world settings, https://arxiv.org/abs/1911.11132

    - 2022, Vim: Out-of-distribution with virtual-logit matching, https://arxiv.org/abs/2203.10807

    - 2022, a review: https://arxiv.org/pdf/2110.11334.pdf


In short, it seems like good work, but it seems like authors know more than the readers (about task definitions, datasets, why newer baselines were not used). Please clarify. Glad to increase score if sufficient answers given to these concerns. Most concerns can be answered without experiments except the "old baselines" concern, which requires either experiments or a commitment to include newer baselines.

**Limitations:**

Yes.

---

> ### Author Rebuttal · Authors · 2023-08-09
>
> We thank the reviewer for the review and constructive feedback.
>
>
> We would like to first clarify that our primary goal is not to perform OOD detection but to understand which concepts a model can and cannot represent (i.e., probing). We designed GPP (Gaussian process probes) to measure uncertainty of “concepts” by using activations/representations as a medium -- a new way to "probe". Probing and delineating different kinds of uncertainty for deep neural net representations has not been well studied, but both fuzziness in concepts and unsureness are ubiquitous cognitive phenomena. When learning new concepts, it is natural for people to perform OOD detection (i.e., more colloquially, being able to say “I’m not sure since I haven’t learned it yet”). GPP fills in the gap between existing probing literature and a person’s intuitive understanding of concepts. This is an important step to advance the new capabilities of explainable and interpretable AI.
>
>
> > “precise definition” and “citation for LPE” and “distributions” in LPE
>
>
> Thank you for pointing out this omission. Linear probe ensembling (LPE) adopts a standard bootstrap aggregating method to ensemble linear classifiers. The same method was used as a component in Kim et al., 2018. Tran et al., 2022 used ensembling and an entropy score for OOD detection. We will include both citations.
>
>
> Precise description of LPE: LPE is an ensemble of linear probes. For each linear probe, we train a logistic regression classifier on a dataset sampled from the original set of observations with replacement. The size of the dataset is the same as the number of observations. Each ensemble has 100 linear probes.
>
>
> By bootstrapping, each linear probe in LPE can be viewed as i.i.d. samples from an underlying distribution over classifiers (denoted as $g$ in the paper). Since variance/entropy-based metrics etc are computed with classifier samples, we can use those linear classifiers in LPE to compute those metrics in the same way as GPP (Section 2.4).
>
>
>
>
> > “writing style suggestion”
>
>
> Thank you. We will polish the writing accordingly.
>
>
>
>
> > “Task issue 1” on names of datasets and what is in distribution
>
>
> We thank the reviewer for their comment, and will clarify these points in the paper. The synthetic dataset is the 3D Shapes dataset [Burgess and Kim, 2018]. The ID examples for ImageNet are images that come from the same distribution that the probe observes. For example, consider a probe trained to classify "dogs" vs "cats"; images of dogs and cats would be ID, whereas random noise would be OOD.
>
>
>
>
> > “Task issue 2”: “How are the pre-trained classifiers use?”
>
>
> The pre-trained models are off-the-shelf ImageNet classifiers trained on all 1000 fine-grained classes (e.g., "hummingbird"). The probe takes as input the model's intermediate activations (features), and is trained to classify between two coarse-grained classes (e.g., "bird" vs "cat"). These coarse-grained classes are constructed using the WordNet hierarchy (https://wordnet.princeton.edu/).
>
>
>
>
> > “Did you specifically pull checkpoints that were also trained on the binarized problems?”
>
>
> No, as mentioned above, we used off-the-shelf ImageNet classifiers trained on all 1000 fine-grained classes. These classifiers are the pre-trained models used for probing.
>
>
>
>
> > “did you somehow pool together a multiclass classifier's probabilities for the underlying model + renormalize? What happened to the probabilities assigned to the other classes?”
>
>
> No, the probe used the activations of the pre-trained model. Our goal was to see if the probe can distinguish between ID/OOD data so that it is a reliable probe. We did not need the probability outputs of the original pre-trained model for our purpose.
>
>
>
>
> > “is the out-distribution in both datasets just uniform noise, or uniform noise added to ID data?”
>
>
> The OOD data in both datasets is pixel-wise uniform noise, not a noisy version of the ID data.
>
>
>
>
> > “OOD detection baselines”
>
>
> For OOD detection baselines, we included MSP (maximum predicted softmax probabilities using LP) [Hendrycksand Gimpel, 2016], Maha (negative Mahalanobis distance-based score) [Lee et al., 2018] (this is the 2nd paper pointed out by the reviewer) and LPE (negative predicted variance from linear probe ensembles) [Tran et al., 2022, Kim et al., 2018]. As shown in the extensive analyses of OOD detection methods and tasks in Appendix E of Tran et al., 2022, Maha and LPE (LPE is equivalent to their “Entropy” method for ensembles) achieved top performance, surpassing more recent Ren et al., 2021.
>
>
>  We included preliminary results on “deep nearest neighbors” [Sun et al., 2022] pointed out by the reviewer in the PDF, and we will add it to the revised version of the paper.
>
>
> We thank the reviewer again for the helpful suggestions and we will address those in detail in our revised version.
>
>
>
>
> *References*
>
>
> Been Kim, Martin Wattenberg, Justin Gilmer, Carrie Cai, James Wexler, Fernanda Viegas, et al.Interpretability beyond feature attribution:  Quantitative testing with concept activation vectors(TCAV). InInternational Conference on Machine Learning (ICML), 2018.
>
>
> Dustin Tran, Jeremiah Liu, Michael W Dusenberry, Du Phan, Mark Collier, Jie Ren, Kehang Han,Zi Wang, Zelda Mariet, Huiyi Hu, et al.  Plex: Towards reliability using pretrained large modelextensions.arXiv preprint arXiv:2207.07411, 2022.
>
>
> Jie Ren, Stanislav Fort, Jeremiah Liu, Abhijit Guha Roy, Shreyas Padhy, and Balaji Lakshminarayanan. A simple fix to mahalanobis distance for improving near-ood detection. arXiv preprint arXiv:2106.09022, 2021.
>
>
> Kimin Lee, Kibok Lee, Honglak Lee, and Jinwoo Shin. A simple unified framework for detecting out-of-distribution samples and adversarial attacks. In Advances in Neural Information Processing Systems (NeurIPS), 2018.
>
>
> Yiyou Sun, Yifei Ming, Xiaojin Zhu, and Yixuan Li. Out-of-distribution detection with deep nearest neighbors. InInternational Conference on Machine Learning (ICML), 2022.

---

### Official Review · Reviewer_5Gmc · 2023-07-27

**Soundness:** 3 good
**Presentation:** 2 fair
**Contribution:** 3 good
**Rating:** 6
**Confidence:** 3

**Summary:**

The authors introduce Gaussian process probes, a probabilistic probing method. They use this method to obtain additional insights into the internals of deep learning models, using the concepts of aleatoric and epistemic uncertainty.

**Strengths:**

* This is an interesting and useful addition to the literature on probing methods. It enables obtaining insights into the function of a deep learning model that could not have been obtained before.
* The method is novel to the best of my knowledge.
* The method is well-presented. The methods section is well-structured and clear.

**Weaknesses:**

* Other parts of the paper could use from better writing. In particular, I thought the intro and the experiments section could use additional polishing.
* The paper could benefit from more extensive experiments. At the very least, it would be standard to test the method on more than one dataset.

**Questions:**

* What other methods and datasets could serve as benchmarks and how would the method perform there

**Limitations:**

No concerns about addressing limitations

---

> ### Author Rebuttal · Authors · 2023-08-09
>
> We thank the reviewer for the positive feedback.
>
>
> > “writing”
>
>
> Thank you for these suggestions. We will polish both the intro and experiment sections.
>
>
>
>
> > “test the method on more than one dataset”
>
>
> In the paper, we conducted experiments on 2 standard datasets and 1 set of photographic images for demo: (1) 3D Shapes [Burgess and Kim, 2018], (2) ImageNet [Russakovsky et al., 2015], and (3) online or proprietary real world images. For (1), we constructed **3 datasets based on concept ontology of the 3D Shapes dataset** to set up experiments where we can train several CNN models to get representations for specific tasks. The 3D Shapes dataset helps us to validate the method with a full control over the ground truth labels (with different ontologies) and levels of label noise. The results are shown in Figure 4-7. For (2), we constructed **10 datasets defined by coarse-grained labels of the ImageNet dataset** (Section C.2) to verify the usefulness of epistemic uncertainty predictions from GPP, and performed experiments on OOD detection tasks. The results can be found in Figure 7. For (3), we collected **online and proprietary images** (that the models have never been trained on) to demonstrate the predictions from GPP. The demos can be found in the introduction, Figure 3 and Table 1. Hence in total, we have 14 different settings in which we evaluated the approach, using several different datasets. We will clarify this point in the paper.
>
>
>
>
> > “What other methods and datasets could serve as benchmarks and how would the method perform there”
>
>
> As mentioned above, we used 3 types of datasets (14 settings in total) and we will make this clearer in the paper. For baseline methods, we included established SOTA or near-SOTA methods, including LP (linear probes), SVM probes, LPE (linear probe ensembles) [Kim et al., 2018], MSP (maximum predicted softmax probabilities using LP) [Hendrycksand Gimpel, 2016], and Maha (negative Mahalanobis distance-based score) [Lee et al., 2018]. As shown in the extensive analyses of OOD detection methods and tasks in Appendix E of Tran et al., 2022, Maha and LPE (LPE is equivalent to their “Entropy” method for ensembles) achieved top performance, surpassing more recent proposals from Ren et al., 2021. We will include more OOD detection baselines such as “deep nearest neighbors” [Sun et al., 2022] in the revised version of the paper. Preliminary results on “deep nearest neighbors” can be found in the PDF from the “global” reply.
>
>
>
>
> We thank the reviewer again for the helpful suggestions and we will address them in our paper accordingly.
>
>
> *References*
>
>
> Chris Burgess and Hyunjik Kim. 3D Shapes dataset. https://github.com/deepmind/3dshapes-dataset/,2018.
>
>
> Olga Russakovsky, Jia Deng, Hao Su, Jonathan Krause, Sanjeev Satheesh, Sean Ma, Zhiheng Huang,Andrej Karpathy, Aditya Khosla, Michael Bernstein, Alexander C. Berg, and Li Fei-Fei. ImageNetLarge Scale Visual Recognition Challenge.International Journal of Computer Vision (IJCV), 115(3):211–252, 2015.
>
>
> Been Kim, Martin Wattenberg, Justin Gilmer, Carrie Cai, James Wexler, Fernanda Viegas, et al.Interpretability beyond feature attribution:  Quantitative testing with concept activation vectors(TCAV). InInternational Conference on Machine Learning (ICML), 2018.
>
>
> Dustin Tran, Jeremiah Liu, Michael W Dusenberry, Du Phan, Mark Collier, Jie Ren, Kehang Han,Zi Wang, Zelda Mariet, Huiyi Hu, et al.  Plex: Towards reliability using pretrained large modelextensions.arXiv preprint arXiv:2207.07411, 2022.
>
>
> Jie Ren, Stanislav Fort, Jeremiah Liu, Abhijit Guha Roy, Shreyas Padhy, and Balaji Lakshminarayanan. A simple fix to mahalanobis distance for improving near-ood detection. arXiv preprint arXiv:2106.09022, 2021.
>
>
> Yiyou Sun, Yifei Ming, Xiaojin Zhu, and Yixuan Li. Out-of-distribution detection with deep nearest neighbors. InInternational Conference on Machine Learning (ICML), 2022.

---

> > ### Comment · Reviewer_5Gmc · 2023-08-16
> >
> > Thank you, I acknowledge having read the response. I continue to support acceptance of the paper. I do think the paper could benefit from an improved presentation of the results (e.g., the figure that shows ImageNet results does not include the word "ImageNet" as far as I can tell).

---

### Author Rebuttal · Authors · 2023-08-09

We are encouraged that the reviewers found our work novel (**Reviewers 5Gmc, VzJz, zzXk**), significant (**3D6e**), well-motivated (**iFt6, VzJz**), well-written and easy to follow (**zzXk, 3D6e**). Moreover, **Reviewer 5Gmc** acknowledged that our measures of uncertainty for probing are interesting and useful, and enable “obtaining insights into the function of a deep learning model that could not have been obtained before”. We are also pleased that reviewers recognized that our method was well-presented (**5Gmc**), clear (**iFt6, VzJz**) and solid (**zzXk**).

In the attached PDF, we included the following figures:
1. Figure 1 (for Reviewers iFt6, 5Gmc) presents additional results on OOD detection with a recent baseline method [Sun et al., ICML 2022] suggested in iFt6.
2. Figure 2 (for Reviewer zzXk) shows that LPE is a valid probing method and supports the fact that it was properly trained.

We really appreciate the suggestions and questions from the reviewers, and we reply to them individually below. We will incorporate all feedback in the new version of the paper.

---

### Decision · Program_Chairs · 2023-09-21

**Decision:**

Accept (poster)

**Comment:**

The consensus after the discussion period is this submission should be accepted because of its technical contributions, potential practical impact and presentation clarity. Some reviewers requested changes to writing styles and additional OOD baselines. Would be great if the authors could address these for the camera-ready version.